# Preserved neural dynamics across animals performing similar behaviour

Mostafa Safaie[1,6], Joanna C. Chang[1,6], Junchol Park[2], Lee E. Miller[3], Joshua T. Dudman[2], Matthew G. Perich[4,5,7 ✉] & Juan A. Gallego[1,7 ✉]

Animals of the same species exhibit similar behaviours that are advantageously adapted to their body and environment. These behaviours are shaped at the species level by selection pressures over evolutionary timescales. Yet, it remains unclear how these common behavioural adaptations emerge from the idiosyncratic neural circuitry of each individual. The overall organization of neural circuits is preserved across individuals[1] because of their common evolutionarily specified developmental programme[2–4]. Such organization at the circuit level may constrain neural activity[5–8], leading to low-dimensional latent dynamics across the neural population[9–11]. Accordingly, here we suggested that the shared circuit-level constraints within a species would lead to suitably preserved latent dynamics across individuals. We analysed recordings of neural populations from monkey and mouse motor cortex to demonstrate that neural dynamics in individuals from the same species are surprisingly preserved when they perform similar behaviour. Neural population dynamics were also preserved when animals consciously planned future movements without overt behaviour[12] and enabled the decoding of planned and ongoing movement across different individuals. Furthermore, we found that preserved neural dynamics extend beyond cortical regions to the dorsal striatum, an evolutionarily older structure[13,14]. Finally, we used neural network models to demonstrate that behavioural similarity is necessary but not sufficient for this preservation. We posit that these emergent dynamics result from evolutionary constraints on brain development and thus reflect fundamental properties of the neural basis of behaviour.

The behaviour of each individual in a species is driven by the coordinated activity of neural populations throughout the brain. This activity emerges from the latent dynamics, which are the time-dependent activation of the dominant patterns of neural covariation[9,11]. These latent dynamics seem to be shaped by circuit and biophysical constraints[5–8]. Given the large differences in brain circuitry across individuals from the same species—including in some type-specific neurons, dendritic morphology and receptor distribution[15–19]—it remains unclear how similar adaptive behaviours emerge from such idiosyncratic neural circuitry. One possibility is that unique circuits in each individual generate unique latent dynamics that produce the same behavioural output. Indeed, the high degrees of freedom of neural activity relative to behaviour[20,21] could allow distinct latent dynamics to produce similar behaviour. Alternatively, the same behaviour performed by two individuals could be produced by preserved latent dynamics. This preservation would emerge from the common organization of neural circuits across individuals resulting from a species-specific developmental programme.

Here, we adopt the last hypothesis: different individuals from the same species engaged in the same behaviour generate preserved latent dynamics. We posit that preserved circuit constraints give rise to a species-wide neural landscape and the individual-specific latent dynamics observed during a behaviour are different instantiations of a common trajectory through this landscape (Fig. 1). Our hypothesis provides several testable predictions. First, because low-level details of neural circuits are idiosyncratic, they should not be necessary to account for the emergence of species-typical behaviours. Accordingly, different animals of the same species engaged in the same behaviour should exhibit preserved latent dynamics. Second, the extent of preservation of the latent dynamics across individuals should be constrained by the similarity of the behavioural output. Third, because low-dimensional latent dynamics are found throughout the brain, not just in cortical regions[22–24], we should also observe preserved latent dynamics in structures that have co-evolved with cortex for hundreds of millions of years such as the basal ganglia[14]. Fourth, because covert behaviours seem to be mediated by the same neural circuits as overt behaviours[25], we should find shared latent dynamics across animals performing the same cognitive task.

[1]Department of Bioengineering, Imperial College London, London, UK. [2]Janelia Research Campus, Howard Hughes Medical Institute, Ashburn, TX, USA. [3]Departments of Physiology, Biomedical Engineering and Physical Medicine and Rehabilitation, Northwestern University and Shirley Ryan Ability Lab, Chicago, IL, USA. [4]Département de Neurosciences, Faculté de Médecine, Université de Montréal, Montreal, Quebec, Canada. [5]Mila, Quebec Artificial Intelligence Institute, Montreal, Quebec, Canada. [6]These authors contributed equally: Mostafa Safaie, Joanna C. Chang. [7]These authors jointly supervised this work: Matthew G. Perich, Juan A. Gallego. ✉e-mail: matthew.perich@umontreal.ca; jgallego@imperial.ac.uk

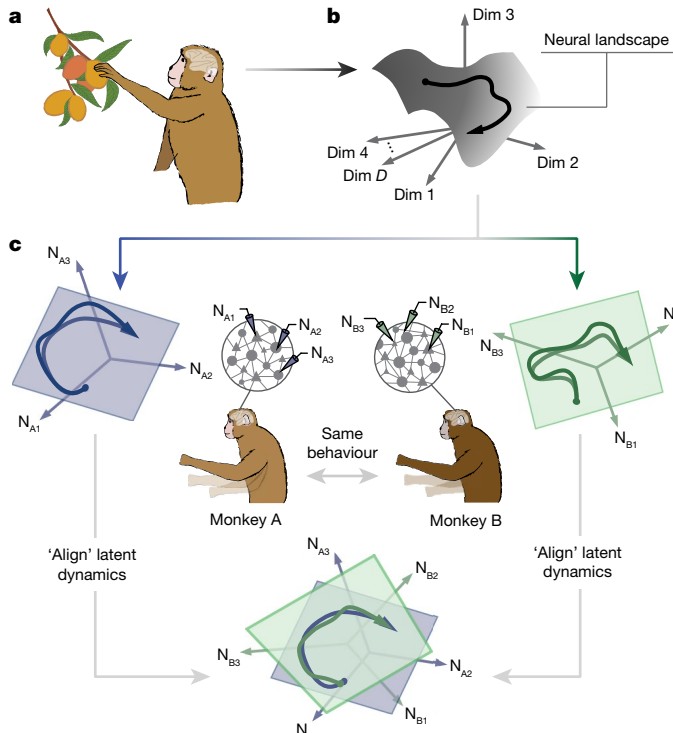

**Fig. 1 | Hypothesis. a,b,** Different individuals from the same species performing the same behaviour will generate preserved neural population latent dynamics by instantiating a species-wide 'neural landscape' embedded in $D$ dimensions (Dim) of neural activity. **c,** These preserved latent dynamics can be revealed by 'aligning' the latent dynamics estimated from neural population recordings of each individual ($N_{A1-3}$ and $N_{B1-3}$ illustrate three neurons recorded from monkey A and monkey B, respectively).

## Preserved latent dynamics across animals

We tested the four predictions outlined above using neural population recordings from monkeys and mice as they performed upper limb tasks. First, we analysed motor cortical recordings from monkeys engaged in an instructed-delay centre-out reaching task with eight targets (Fig. 2a,b and Extended Data Fig. 1a; Methods). All three monkeys were well-trained in the task and exhibited highly stereotyped hand trajectories (mean trajectory correlation for each monkey: $r = 0.89$, 0.90 and 0.92; Extended Data Fig. 1b). For each session, we used principal component analysis (PCA) to estimate the latent dynamics underlying overt movement execution by projecting the firing rates of each recorded neuron (or multi-unit, for monkey J) onto the leading ten PCA axes (the neural modes; examples in Fig. 2c; Extended Data Fig. 1c). We then aligned the latent dynamics of each pair of experimental sessions from two different animals using canonical correlation analysis (CCA[26]), a method that maximizes the correlations between two sets of signals through linear transformations (similar to refs. 26–28).

This linear method revealed that the ostensibly different latent dynamics of two different monkeys are indeed highly preserved (Fig. 2e). The across-animal correlations greatly exceeded two lower-bound controls. The first was established by aligning randomly selected behavioural epochs ('control' in Fig. 2f) and the second was based on surrogate data that conserved the statistical structure of the neural activity[29] (tensor maximum entropy (TME); Extended Data Fig. 2a,b). Most importantly, these correlations were nearly as high as values obtained by aligning two subsets of trials within a single session from the same animal ('within' in Fig. 2f; further examples in Extended Data Fig. 3a). This result further held across all pairs of sessions from all three monkeys ($n = 126$ sessions; Fig. 2g). The aligned neural modes

captured a large fraction of neural variance (Extended Data Fig. 2e) and the results did not depend on the assumed dimensionality of the neural manifold (Extended Data Fig. 2f) or the alignment method (Extended Data Fig. 2g,h).

Although we have demonstrated the preservation of latent dynamics across animals, these shared dynamics may not necessarily be relevant to behaviour. To address this, we trained neural network decoders[30] (long short-term memory networks (LSTMs)) to predict the hand trajectories of one animal and tested their performance on a second animal. The across-animal decoding accuracy approached the upper bound provided by the performance of decoders trained and tested on the same animal (Fig. 2h; example predictions in Extended Data Fig. 4). Thus, the preserved motor cortical dynamics across animals contain detailed information about the ongoing movement kinematics.

We then analysed data from four mice trained to perform a reaching and pulling joystick task (Fig. 2i and Extended Data Fig. 5). We found that both within and across individuals, the behavioural output was less similar from trial to trial than for the monkey dataset (compare Extended Data Figs. 1a,b to 5a,b). As predicted, our alignment procedure revealed that motor cortical latent dynamics were largely preserved across mice both when reaching to two different targets (Fig. 2j and Extended Data Figs. 3b and 2c; example in Extended Data Fig. 3e) and when subsequently pulling at two different force levels (Extended Data Fig. 6a). Yet, these correlations were lower than those of monkeys (compare Fig. 2g to 2j), which directly impacted our ability to decode movement kinematics across animals (Extended Data Fig. 6b). This difference could be explained by the more highly stereotyped behaviour of the monkeys compared to the mice (Fig. 2k inset). Comparing between species confirmed that behavioural stereotypy was associated with both the preservation of the latent dynamics across individuals (Fig. 2k) and across-animal decoding accuracy (Extended Data Fig. 6f). These results demonstrate in two evolutionarily divergent species that there is a direct correspondence between the similarity of behavioural output and the preservation of motor cortical latent dynamics across individuals.

## Necessity of behavioural similarity

In the preceding analyses, we studied tasks comprising a few conditions that inadvertently imposed a topological structure in the produced movements[12,26]. We sought to establish that preserved latent dynamics do not merely reflect this structure. First, we tested whether this topological structure is sufficient to produce preserved latent dynamics and found that the preservation across individuals was significantly impaired (Extended Data Fig. 7). We then further demonstrated the existence of preserved latent dynamics during a continuous and less-structured task in which monkeys rapidly generated sequences of random reaches[31,32] (Fig. 3a). The produced movements were highly varied with little organization (Fig. 3b and Extended Data Fig. 8a). To facilitate alignment, we parcelled the workspace to match movements across individuals on the basis of the initial hand position and reach direction to generate as many as 29 similar reaching conditions (Fig. 3c). Despite the dramatic increase in number of conditions and behavioural variability, we could still uncover preserved latent dynamics across individuals (Fig. 3d,e and Extended Data Fig. 2d). The increased complexity of behavioural output in this task allowed us to directly study the relationship between the number of conditions and the preservation of latent dynamics. We subsampled the conditions and found that latent dynamics across individuals were preserved for the entire range (up to 29) considered (Fig. 3f and Extended Data Fig. 8b) even when as few as about 20 neurons are included (Fig. 3g and Extended Data Fig. 8c). The preservation of latent dynamics across individuals decreased when we shuffled the conditions (Extended Data Fig. 8d), thereby reducing behavioural similarity. This agrees with

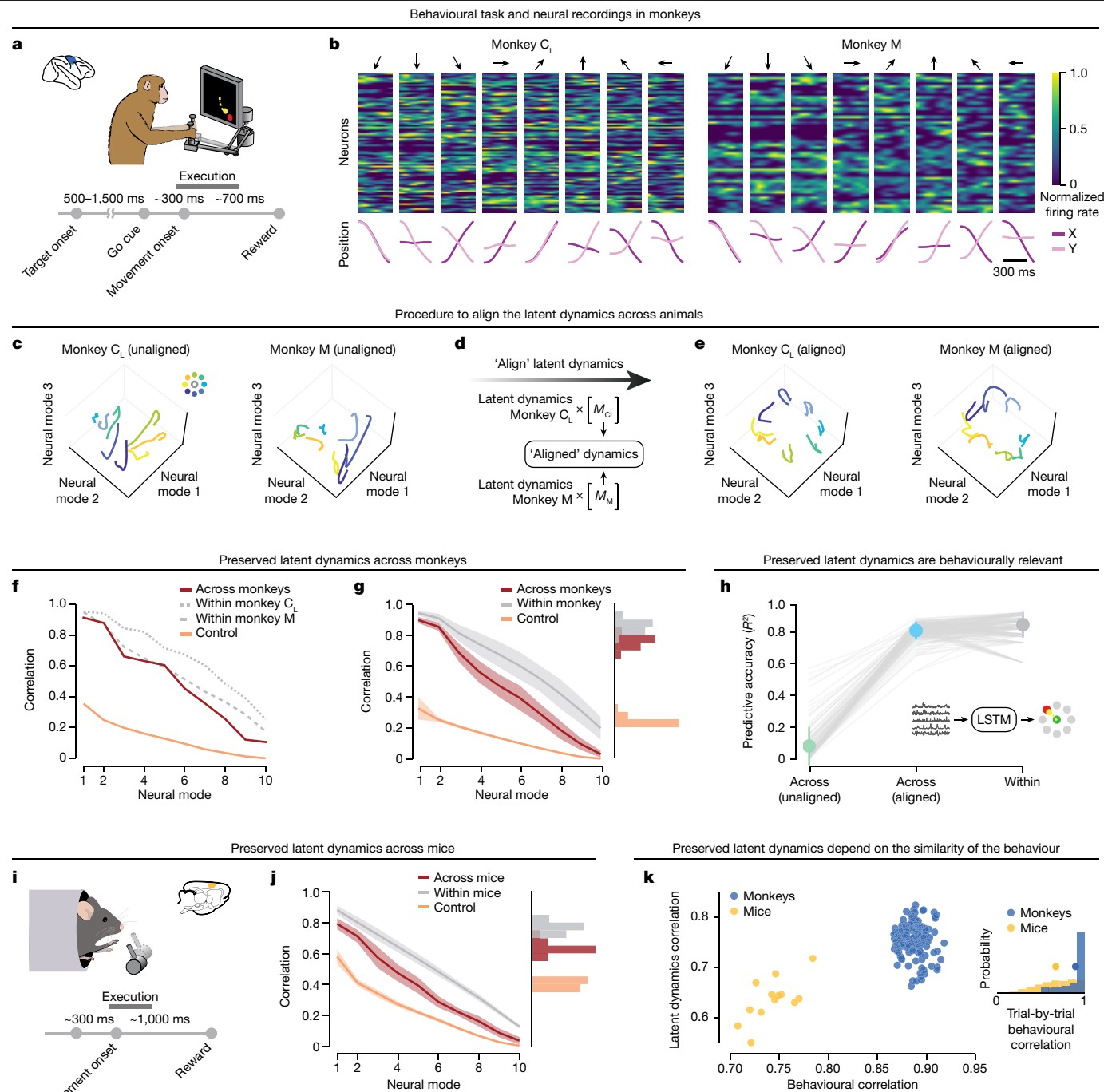

**Fig. 2 | Preserved latent dynamics across animals performing the same behaviour. a**, Monkeys performed an eight-target instructed-delay centre-out reaching task using a planar manipulandum. **b**, Example normalized neural firing rates aligned to movement onset (top) and hand trajectories (bottom) for two monkeys. Each column shows one reach to the eight targets indicated by the arrows. Note that monkey C received two sets of motor cortical implants, $C_L$ and $C_R$, with $C_L$ denoting the implant in the left hemisphere and $C_R$ the implant in the right hemisphere. **c–e**, Three-dimensional representation of the motor cortical latent dynamics for the two monkeys plotted in **b** before (**c**) and after (**e**) alignment with CCA (**d**). **f**, Correlations of the aligned (red) latent dynamics for the example comparison in **c–e** compared to within-monkey correlations (grey) and a lower-bound control (orange). **g**, Preserved latent dynamics across all pairs of 21 sessions ($n = 126$ comparisons) from three different monkeys. Histograms show the mean correlation across the leading four dimensions. Line and shaded area, mean ± s.d. **h**, Decoders trained on aligned latent dynamics from one monkey predict continuous hand kinematics of a different

monkey (blue). Results compared to decoders trained and tested within the same session (grey) and without alignment (green). Data points, individual comparisons ($n = 126$) between sessions from different monkeys. Error bars, mean ± s.d. Statistical tests: two-sided Wilcoxon's rank sum test, $P = 3.1 \times 10^{-6}$ for decoding performance between across-animal correlation and within-animal correlation, $P = 2.0 \times 10^{-22}$ between across-animal correlation and lower bound. **i**, Mice grasped and pulled a joystick in two positions (left or right). **j**, Preserved latent dynamics across mice performing the grasping and pulling task. Data include six sessions across four different mice ($n = 13$ comparisons), formatted as in **g**. **k**, Preserved latent dynamics across animals is related to the similarity of their behaviour. For each session pair, the mean of the top four canonical correlations (CCs) between the latent dynamics against the mean behavioural correlation. Single dots, pairs of sessions colour-coded by species. Inset: behavioural correlations for all pairs of trials from different mice and monkeys. Circles, mean.

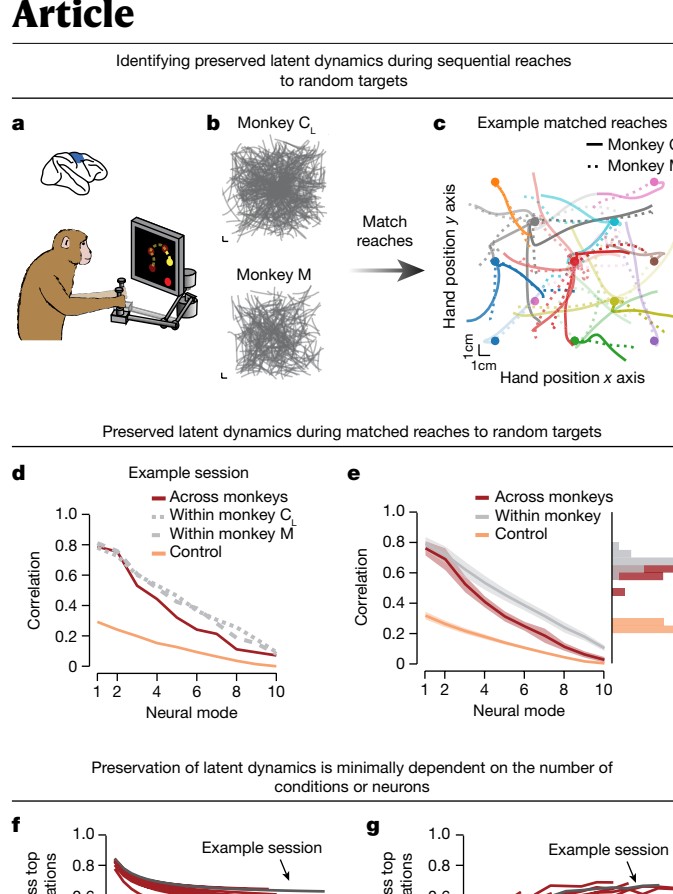

Identifying preserved latent dynamics during sequential reaches to random targets

**a**

**b** Monkey $C_L$

Monkey M

Match reaches →

**c** Example matched reaches
— Monkey $C_L$
···· Monkey M

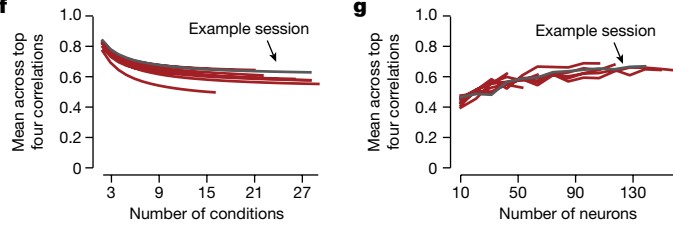

Preserved latent dynamics during matched reaches to random targets

**d** Example session
— Across monkeys
···· Within monkey $C_L$
--- Within monkey M
— Control

**e**
— Across monkeys
— Within monkey
— Control

Preservation of latent dynamics is minimally dependent on the number of conditions or neurons

**f** Example session

**g** Example session

**Fig. 3 | Preserved latent dynamics in a sequential reaching task with higher behavioural complexity. a**, Monkeys were trained to perform sequences of four reaches to randomly placed target locations using the planar manipulandum. **b**, Example hand positions in the workspace for all reaching movements made during a session from two example monkeys. **c**, We matched similar movements between the two monkeys on the basis of their starting location, duration and direction. Each colour represents a different representative condition and matched reaching trajectories are overlaid for the two monkeys. Circles, starting position. **d**, Preserved motor cortical latent dynamics across monkeys performing matched reaches for the example sessions in **c**. Note that the correlation between aligned latent dynamics across monkeys was quite close to the within-monkey correlations (grey) and largely exceeded our lower-bound control (orange). **e**, Preserved motor cortical latent dynamics across all pairs of sessions from three different monkeys. Data include $n = 10$ comparisons and are presented as in Fig. 2g. Line and shaded area, mean ± s.d. **f**, Preserved latent dynamics hold for a wide range of conditions. The plot shows, for each pair of sessions, the mean of the top four CCs between the latent dynamics as a function of the number of conditions subsampled from the total available in each session. Dark trace, example session from **b**–**d**. **g**, Preserved latent dynamics can be uncovered from even sparsely sampled neural populations. The plot shows, for each pair of sessions, the mean of the top four CCs between the latent dynamics as a function of the number of neurons subsampled from the total available in each session. Dark trace, example session from **b**–**d**.

the previous comparison of mice and monkeys: monkeys performing a more complex task (eight conditions) than mice (two to four conditions) had higher preservation of latent dynamics because of the more stereotyped behavioural output.

We performed more control analyses to confirm that this relationship between behavioural stereotypy and preservation of latent

dynamics across individuals from the same species is not a trivial consequence of our methodology. First, we compared our results with a previous study that investigated the dynamics of motor cortical activity within an individual across different but related wrist manipulation and reach-to-grasp tasks[33] (Extended Data Fig. 9). During each task, individuals activated the same muscles in a slightly different manner (Extended Data Fig. 9b–d). Accordingly, the latent dynamics of the same monkey performing two distinct but related behaviours were much less preserved than those of different monkeys performing the same behaviour (Extended Data Fig. 9g). Last, as we have previously shown, preserved latent dynamics cannot be explained by stable movement tuning[26], nor do they persist following nonlinear transformations[26]. These analyses demonstrate that the alignment method alone is not sufficient to uncover preserved latent dynamics, even within individuals.

## Preserved dynamics in dorsal striatum

Given that the motor cortex is the main cortical output to the spinal circuits that generate movement, the close correspondence between motor cortical latent dynamics and behavioural output may uniquely result from the architecture and projections of this region. To test whether preserved latent dynamics exist across the brain, we studied the subcortical nuclei of basal ganglia, which do not directly project to spinal cord but are crucial for various aspects of behaviour[34–38]. We predicted that basal ganglia latent dynamics would also be preserved across animals performing the same task. Replicating our alignment analysis on neural population recordings from mouse dorsolateral striatum during a reaching and pulling task (Fig. 4a) revealed preserved latent dynamics across individuals (Fig. 4b and Extended Data Fig. 3c; example in Extended Data Fig. 3f).

Moreover, despite the vast differences in circuit and cellular architecture between motor cortex and striatum[38,39], both the across-animal correlations (compare Fig. 4b and Fig. 2j) and the across-animal decoding performance of hand trajectories (Fig. 4c and Extended Data Fig. 6f) were equally large for both regions (Extended Data Fig. 6c–e). Therefore, stable latent dynamics across animals performing the same behaviour are not confined to motor cortex—they extend to different regions throughout the brain, including an evolutionarily older structure that is shared among all vertebrates[13].

## Preserved dynamics during covert behaviour

We have shown the preservation of latent dynamics across brain regions during active, overt behaviour. However, animals also engage in a variety of covert behaviours such as deliberation and planning. These processes require neural activity that is predominantly internally generated by the brain. Given that such covert behaviours involve brain regions that also mediate overt behaviours[25,40], we predicted that the latent dynamics underlying these more cognitive processes would also be shared across individuals of the same species. We tested this prediction by analysing motor cortical activity as monkeys planned an upcoming movement before executing it (Fig. 4d). The latent dynamics during the instructed-delay period were highly correlated across animals (Fig. 4e and Extended Data Fig. 3d; example in Extended Data Fig. 3g) and were virtually identical to those during overt reaching behaviour (compare Fig. 2g). Moreover, these aligned latent dynamics were also predictive of behaviour: Bayesian models predicting the upcoming reaching target based on the aligned latent dynamics from one monkey generalized to another (Fig. 4f). Thus, different individuals use preserved latent dynamics not only to execute the same movement but also to perform the same covert mental process. This result also strengthens our previous observation of preserved latent dynamics during overt behaviour. Afferent feedback arriving

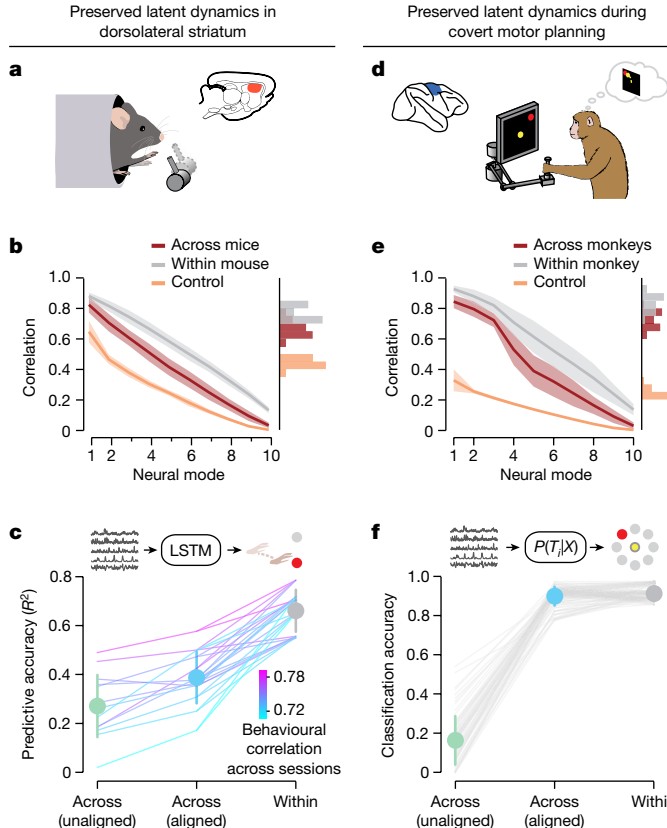

**Fig. 4 | Preserved latent dynamics in the basal ganglia and during covert behaviour. a**, We studied whether shared latent dynamics can also be found in subcortical regions by analysing recordings from mouse dorsolateral striatum during a reaching and pulling task. **b**, Preserved striatal latent dynamics across mice performing the same grasping and pulling task. Data include $n = 13$ comparisons across four mice and are presented as in Fig. 2g. Line and shaded area, mean ± s.d. **c**, Hand trajectories can be decoded across mice using their preserved striatal latent dynamics. Data presented as in Fig. 2g. Error bars, mean ± s.d. Statistical tests: two-sided Wilcoxon's rank sum test, $P = 3.0 \times 10^{-8}$ for decoding performance between across-animal correlation and within-animal correlation, $P = 7.3 \times 10^{-4}$ between across-animal correlation and lower bound. **d**, To investigate the preservation of latent dynamics during covert behaviour, we examined the preparatory period preceding movement execution in the monkey centre-out task. **e**, Preserved motor cortical latent dynamics when preparing to reach to a target. Data include 18 sessions and $n = 72$ comparisons across two different monkeys and are presented as in Fig. 2g. Line and shaded area, mean ± s.d. **f**, Preserved latent dynamics during covert behaviour contain behaviourally relevant information. Naïve Bayes classifiers trained on the aligned latent dynamics (blue) predicted the intended target in a different monkey virtually as well as classifiers trained and tested on the same monkey (grey), whereas classification was poor without alignment (green). Error bars, mean ± s.d. Statistical tests: two-sided Wilcoxon's rank sum test, $P = 1.6 \times 10^{-4}$ for classifier performance between across-animal correlation and within-animal correlation, $P = 1.7 \times 10^{-13}$ between across-animal correlation and lower bound.

at the motor cortex[41,42] could partially explain the observed similarity in latent dynamics during overt movement, yet the latent dynamics are entirely internally generated during covert processes such as movement planning.

## Behavioural similarity is not sufficient

Our hypothesis requires that behavioural similarity is necessary but not sufficient to allow for alignment of latent dynamics. To test this, we trained recurrent neural networks (RNNs) to produce similar

behavioural output while generating distinct latent dynamics. We devised an RNN simulation in which we had control over the degree of preservation of latent dynamics by varying a parameter of the cost function, $\alpha$ (Fig. 5a and Extended Data Fig. 10). We thus created pairs of models that generated highly similar behaviour (Fig. 5b–d) but exhibited distinct latent dynamics as evidenced by the relative lack of alignment (Fig. 5e). We predicted that this decrease in preservation of latent dynamics would be driven by differences in underlying circuit properties. When we reverse-engineered the weights of the different networks, we found that more dissimilar latent dynamics corresponded to larger changes in the variance and dimensionality of the weights changes during training (Fig. 5f). Thus, preservation of latent dynamics is not just a trivial consequence of behavioural similarity; instead, it probably reflects fundamental organization and constraints in the underlying circuit implementation.

## Discussion

Neural population latent dynamics have been proposed as first-level explainers of behavioural and cognitive functions[9,11], a view that has shed light onto the neural basis of numerous phenomena, such as processes underlying covert[25,40,43] and overt behaviour[33,44,45], how learning may happen in neural circuits[5,46,47] and how information may flow between different brain regions[42,48,49]. Here, we extend recent works[20,50–52] to show that latent dynamics are shared across different individuals engaged in the same behaviour for a range of behavioural complexity. The discovery of preserved latent dynamics across individuals will impact both fundamental and applied neuroscience, in particular the development of brain-controlled devices such as neuroprosthetics[53,54]: with proper alignment, decoders trained on one participant could be readily translated to other individuals[55–59] to minimize training and deployment time.

We studied two regions (motor cortex and dorsolateral striatum) whose functions are tightly linked to the production of limb movements. In the behaviours studied here, we demonstrated preserved latent dynamics using a relatively few dimensions. This low dimensionality could be attributed in part to the temporally smooth and relatively constrained movements in these datasets. Further, the limbs of different individuals from the same species share similar biomechanical properties that, throughout evolution and development, have imposed extra constraints enforcing the preservation of latent dynamics. By contrast, highly sensory-driven regions with rapidly changing inputs may require a higher level of granularity in the analysis but we nonetheless expect to see similar preservation of latent dynamics across individuals. 'Higher' brain regions (for example, frontal cortex) that serve more abstract cognitive functions may show less preservation of latent dynamics across animals because of differences in their internal states (attention, motivation, satiation and so on). Even if the most task-relevant aspects of the latent dynamics were preserved across individuals, the influence of these more abstract internal states[60,61] could alter the overall latent dynamics, making the activity of these regions less amenable to alignment. Moreover, different animals may also use different covert strategies to solve the same cognitive task based on their biases acquired through learning and past experience[62]. Yet, we expect that the neural population latent dynamics would still be preserved if it were possible to appropriately match the internal states or strategies across individuals.

The influence of learning and experience on neural circuit organization and the resulting latent dynamics is a fascinating open question. Throughout an individual's lifetime, new skills are acquired through practice, which are consolidated through changes in neural circuitry[63,64]. What, if any, differences should exist in the latent dynamics produced by the motor cortices of, say, a virtuoso guitarist and a first-year guitar student? Regardless of whether those circuits were shaped by practice or even development, we propose that neural circuits are tuned to

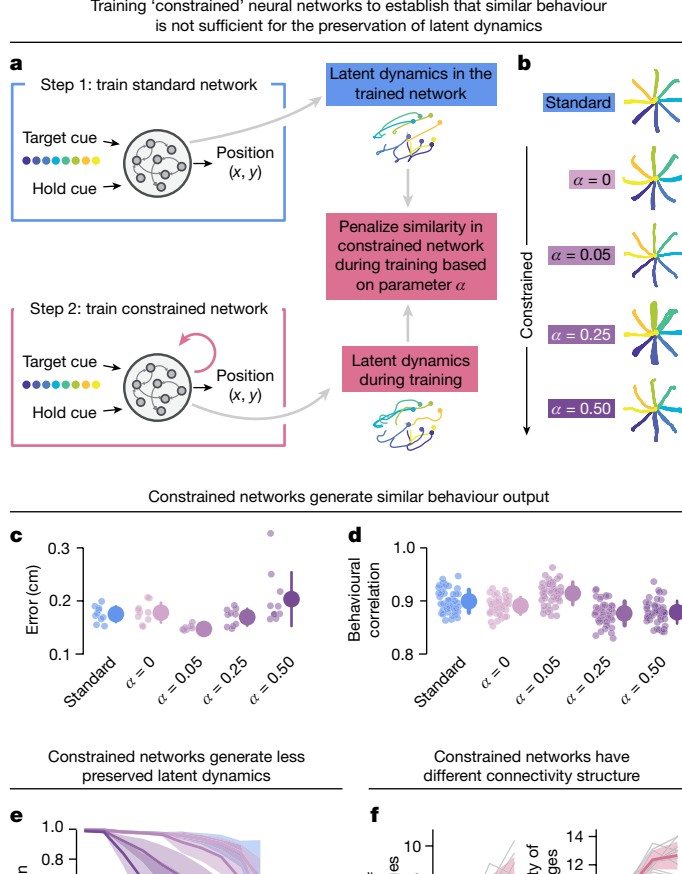

Training 'constrained' neural networks to establish that similar behaviour is not sufficient for the preservation of latent dynamics

**a**

Step 1: train standard network

Target cue → ● ● ● ● ● ● → Position (x, y)
Hold cue →

→ Latent dynamics in the trained network

Penalize similarity in constrained network during training based on parameter α

Step 2: train constrained network

Target cue → ● ● ● ● ● ● → Position (x, y)
Hold cue →

→ Latent dynamics during training

**b**

Standard
α = 0
α = 0.05
α = 0.25
α = 0.50

Constrained

Constrained networks generate similar behaviour output

**c** Error (cm): Standard, α = 0, α = 0.05, α = 0.25, α = 0.50

**d** Behavioural correlation: Standard, α = 0, α = 0.05, α = 0.25, α = 0.50

Constrained networks generate less preserved latent dynamics

**e** Correlation vs Neural mode (1 2 4 6 8 10)
— Standard
— α = 0
— α = 0.05
— α = 0.25
— α = 0.50

Constrained networks have different connectivity structure

**f** Variance of weight changes; Dimensionality of weight changes (α = 0, α = 0.05, α = 0.25, α = 0.50)

**Fig. 5 | Similar behavioural output is necessary but not sufficient for the preservation of latent dynamics. a**, We trained two sets of RNN models to perform the monkey centre-out reaching task. The first set of models were standard RNNs, whereas the second were constrained to produce distinct latent dynamics from the standard networks. The weight of this extra constraint was controlled by parameter $\alpha$. **b**, Example movement trajectories for standard and constrained networks. **c,d**, Both standard and constrained networks produced similar behavioural output, as shown by the error (**c**) and the correlation between 'reach trajectories' across networks (**d**). Data show ten networks initialized from different random seeds for each type of network ($n = 45$ comparisons within each network condition). Error bars, mean ± s.d. **e**, Increasing the value of $\alpha$ decreased the preservation of latent dynamics, establishing that behavioural similarity is necessary but not sufficient to have preserved latent dynamics. Line and shaded area, mean ± s.d. **f**, Networks with distinct latent dynamics show differences in underlying connectivity. The variance and the dimensionality of the weight changes increased as the latent dynamics were constrained to be more distinct from those of standard networks by increasing $\alpha$. Variance values are scaled by $10^{-5}$. Line and shaded area, mean ± s.d.

produce latent dynamics as a solution for behavioural output. In this way, preserved latent dynamics across individuals reflect a fundamental property of the neural basis of behaviour.

Here, we introduced two competing possibilities to explain how similar behaviour can emerge from different neural circuitry. In the first hypothesis, each individual's brain must learn during development to produce latent dynamics within their idiosyncratic neural circuits to enable the desired behaviour, with no guarantee that both

individuals arrive at the same 'solution'. In the second hypothesis, the generated latent dynamics for the desired behaviour are constrained by the genetically specified organization at the circuit level shared by each individual of a given species. Our results support this second hypothesis and raise an intriguing possibility: given that in higher vertebrates the genome does not specify the implemented architecture in great detail, for example, to the level of the synapse[2–4], the genome may provide a 'generative model' that is instantiated by each individual's brain. This generative model may constrain low-level details of the neural circuitry such that the appropriate neural population latent dynamics required for the behavioural repertoire emerge throughout development.

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

## Methods

### Subjects and behavioural tasks

**Monkeys.** We trained four monkeys (monkeys C, M and J: male, *Macaca mulatta*; monkey T: male, *M. fascicularis*; aged 6–10 years) to sit in a primate chair and make reaching movements using a customized planar manipulandum. The movement of a cursor on a computer screen was mapped to the motion of the handle of the manipulandum and the behavioural task was run through custom software in Matlab (The Mathworks). Monkeys C, M and J were trained to perform a two-dimensional centre-out reaching task for at least several months before the neural recordings, ensuring they had reached expert performance. Monkeys C, M and T were trained on a more complex random target sequential reaching task. In the centre-out task, the monkey moved its hand to the centre of the workspace to begin each trial. After a variable waiting period, the monkey was presented with one of eight outer targets. The targets were equally spaced in a circle and selected randomly with uniform probability. Then, an auditory go cue signalled the animals to reach to the target. Monkeys were required to reach the target within 1 s after the go cue and hold for 0.5 s to receive a liquid reward, except for monkey J, who was trained without the instructed-delay period or the 0.5 s target hold time and therefore made larger movements (Extended Data Fig. 1a, right). For the centre-out task, there were 12 sessions for monkey C, 6 sessions for monkey M and 3 sessions for monkey J.

In the random target task, the monkeys made four consecutive reaches to random targets within a $10 \times 10$ cm$^2$ workspace in each trial. Each target was presented sequentially in a random location within an annulus with 5 cm inner radius and 15 cm outer radius of the previous target to enforce minimum and maximum reach lengths. Monkeys received a liquid reward during a short break after each successful sequence of four random target acquisitions. There was no explicit auditory go cue and only a brief hold period within the target (100 ms) and then a brief delay period (100 ms) before the next target was presented. These short constraints helped to enforce that the monkeys made separate, directed movements but did not require that the monkey necessarily stop between movements. For the random target task, there was one 'reference' session for monkey C, six sessions for monkey M and four sessions for monkey T. As the monkeys performed these tasks, we recorded the position of the endpoint at a sampling frequency of 1 kHz using encoders in the joints and digitally logged the timing of task events, such as the go cue. Portions of the centre-out reaching data have been previously published and analysed in refs. 26,28,46,65. Portions of the random target data have been previously published and analysed in refs. 31,32.

**Mice.** Four 8–16-week-old mice were trained to perform a forelimb reaching and pulling task (similar to refs. 38,66) for approximately one month, following habituation to head-fixation and the recording setup. In each trial, mice had to reach and pull a joystick positioned about 1.5 cm away from the initial hand position. The joystick appeared, without any cue, in one of two positions (left or right, less than 1 cm apart). Mice could then self-initiate a reach to the joystick and pull it inwards to get a liquid reward. The joystick was weighted with either a 3 or a 6 g load (light or heavy), making up a total of four trial types (two joystick positions by two loads). Each trial type was repeated 20 times before task parameters were switched to the next trial type without any cue. Each session consisted of two repetitions of each set of four trial types presented in the same order, making up $2 \times 4 \times 20 = 160$ trials. Trials with incorrect responses (for example, pushing the joystick past a threshold, 5 mm) or timeout (the lack of pull or push for 10 s) were marked as unsuccessful. All joystick operations were programmatically controlled using a custom-written open-source Python package: (https://github.com/janelia-pypi/mouse_joystick_interface_python). Mice were maintained on a 12/12 h (08:00–20:00) light/dark cycle and recordings were

made between 09:00 and 15:00. The holding room temperature was maintained at $21 \pm 1$ °C with a relative humidity of 30–70%.

There were two sessions for mouse 38, one session for mouse 39, two sessions for mouse 40 and one session for mouse 44. Movement kinematics were tracked using markerless video-based pose estimation. Annotation of behaviour was accomplished using Janelia Automatic Animal Behavior Annotator[67]. Briefly, behaviour was recorded using two synchronized high-speed (500 frames s$^{-1}$), high-resolution monochrome cameras (Point Grey Flea3; 1.3 MP Mono USB3 Vision VITA 1300; Point Grey Research) with 6–15 mm (f/1.4) lenses (C-Mount), placed perpendicularly in front and to the right of the animal. A custom-made near-infrared light-emitting diode light source was mounted on each camera. Video was recorded using custom-made software developed by the Janelia Research Campus Scientific Computing Department and IO Rodeo. This software controlled and synchronized all facets of the experiment. For the main analyses, light and heavy trials were pooled together because we focused on the reaching phase of the task and the location of the joystick does not depend on its weight. Note that in Extended Data Fig. 6a we repeated the main analysis to demonstrate preserved latent dynamics during the pulling phase, considering all four conditions.

### Neural recordings

**Monkeys.** All surgical and experimental procedures were approved by the Institutional Animal Care and Use Committee of Northwestern University under protocol no. IS00000367. We implanted 96-channel Utah electrode arrays in the primary motor cortex (M1) or dorsal premotor cortex (PMd) using standard surgical procedures. Throughout the paper, neural recordings from these two subregions were pooled together and denoted as motor cortex. This allowed us to ensure that we could evaluate overt and covert dynamics within the same population. Implants were done in the opposite hemisphere of the hand animals used in the task. Monkeys M and T received two arrays in M1 and PMd simultaneously. Monkey J received a single array in M1. Monkey C received two sets of implants: one array in the right M1 while performing the task using the left hand and, following removal of this original implant, two arrays simultaneously in the left M1 and PMd while using the right hand (respectively, monkeys $C_R$ and $C_L$ in our previous work[26]). Note that for all across-individual analyses, $C_R$ and $C_L$ are considered the same animal.

Neural activity was recorded during the behaviour using a Cerebus system (Blackrock Microsystems). The recordings on each channel were band-pass filtered (250 Hz–5 kHz) and then converted to spike times on the basis of threshold crossings. The threshold was set to 5.5× the root-mean-square activity on each channel. We also manually spike sorted the recordings from monkeys C, M and T to identify putative single neurons. Monkey J had fewer well-isolated single units than the other monkeys, so rather than spike sorting we directly applied the multi-unit threshold crossings acquired on each electrode. However, it has been shown that the latent dynamics estimated from multi-unit and single neuron activity are similar[68], an observation that holds true for aligning latent dynamics with CCA[26] (note that we refer to both single neurons and multi-units simply as units). We included multiple experimental sessions from each monkey: for the centre-out reaching task, eight from monkey $C_L$, four from monkey $C_R$, six from monkey M and three from monkey J (example data in Extended Data Fig. 1); for the random target task, one 'reference session' from monkey C, six from monkey M and four from monkey T (example data in Extended Data Fig. 8). These experimental sessions were chosen on the basis of the high number of units or trials and blind to the behaviour of the animal. For the centre-out reaching task, the average number of units included for each monkey was: monkey $C_L$, $277 \pm 14$ (mean ± s.e.m.; range, 210–345); monkey $C_R$, $85 \pm 4$ (range, 73–92); monkey M, $117 \pm 4$ (range, 106–130); and monkey J, $63 \pm 9$ (range, 54–81). For the random target task, the average number of units included was: monkey $C_L$, 280

(one session only); monkey M, 127 ± 9 (range, 101–153); and monkey T, 49 ± 8 (range, 30–66). A more detailed description of the behavioural and neural recording methods is presented in ref. 26.

**Mice.** All surgical and experimental procedures were approved by the Institutional Animal Care and Use Committee of Janelia Research Campus. A brief (less than 2 h) surgery was first performed to implant a three-dimensional-printed headplate[69]. Following recovery, the water consumption of the mice was restricted to 1.2 ml per day, to train them in the behavioural task. Following training, a small craniotomy for acute recording was made at 0.5 mm anterior and 1.7 mm lateral relative to bregma in the left hemisphere. A neuropixels probe was centred above the craniotomy and lowered with a 10° angle from the axis perpendicular to the skull surface at a speed of 0.2 mm min$^{-1}$. The tip of the probe was located at 3 mm ventral from the pial surface. After a slow and smooth descent, the probe was allowed to sit still at the target depth for at least 5 min before initiation of recording to allow the electrodes to settle.

Neural activity was filtered (high-pass at 300 Hz), amplified (200× gain), multiplexed and digitized (30 kHz) and recorded using the SpikeGLX 3.0 software (https://github.com/billkarsh/SpikeGLX). Recorded data were preprocessed using an open-source software KiloSort 2.0 (https://github.com/MouseLand/Kilosort) and manually curated using Phy (https://github.com/cortex-lab/phy) to identify putative single units in each of the primary motor cortex and dorsolateral striatum. A total of six experimental sessions (from four mice; Extended Data Fig. 5) with simultaneous motor cortical and striatal recordings were included in this work. The average number of motor cortical units included for each mouse was: mouse 38, 98 ± 4 (range, 95–102); mouse 39, 64; mouse 40, 75 ± 5 (range, 70–80); and mouse 44, 55. The average number of striatal units included for each mouse was: mouse 38, 100 ± 13 (range, 87–112); mouse 39, 108; mouse 40, 74 ± 5 (range, 69–79); and mouse 44, 110.

## Data analysis

We used a similar approach for both monkey and mouse data. In all the analyses, we only considered the trials in which the animal successfully completed the task within the specified time and received a reward. We concatenated trials in time for subsequent analyses—that is, no trial-averages were taken. For the monkey centre-out reaching task and the mouse reaching and pulling task, an equal number of trials to each target was randomly selected (eight targets for the monkeys and two targets for mice, except in Extended Data Fig. 6a, for which four targets were considered). Trial order was randomized to eliminate the possible effect of the passage of time. Within each trial, we isolated a window of interest that captured most of the movement, starting 50 ms before movement onset and ending 400 ms after movement onset. To analyse covert behaviour in monkeys, we used a window that spanned the movement planning period, which started 400 ms before movement onset and ended 50 ms after movement onset. Importantly, all of our results held when changing the analysis windows within a reasonable range.

For the monkey random-walk task, each reach could start and end anywhere within the workspace. To define movements (conditions) that could be matched across animals, we first segmented the workspace into 12 circular subsections. Each subsection was then divided into six equal sectors and targets in the same angular sector were grouped together, creating 72 possible target conditions. We separated the sequences of four consecutive reaches and considered each reach as a separate movement. To assign each movement to a target condition, we first assigned each movement to one of the subsections on the basis of the starting position of the given movement, excluding movements that started more than 2 cm from the centre of the subsection. We then recentred the movements so that they started in the centre of each subsection and reached outwards towards their target. The movement was then assigned to a sector and target condition on the basis of the angle

of target. To study the preservation of latent dynamics across monkeys performing similar behaviour, we needed to match movements (reach conditions) across sessions for different monkeys. To maximize the number of matched movements, we compared all sessions for Monkey M and Monkey T against a reference session for Monkey $C_L$ that had the most successful trials. We matched movements in each pair of sessions by minimizing the mean squared error (MSE) between pairs of movements, excluding matches that had MSEs above the threshold of 2% of MSEs calculated for all possible pairs of movements. If the matched movements had different corresponding target conditions, we used the target condition label from the reference session. After this process was completed, we excluded target conditions with less than six matched movements, such that paired sessions had up to 29 shared target conditions. Because these movements were more ballistic than in the centre-out task, we examined a window starting at movement onset and ending 350 ms after movement onset.

All the analyses were implemented in Python using open-source packages such as numpy, matplotlib, sci-kit, scipy and pandas[70–74] and custom code. As we were analysing existing datasets on an individual basis, no explicit planning of sample size, group randomization or blinding was performed.

**Behavioural correlation.** To assess the behavioural stereotypy of a given animal, we calculated hand trajectory correlations (Pearson's correlation) of every pair of trials within a session (Extended Data Fig. 1b and Extended Data Fig. 5b). The distributions in Fig. 2k inset illustrate these correlations pooled across all the monkey centre-out and mouse reaching and pulling sessions included in this work. To determine the behavioural similarity across pairs of sessions from different monkeys or mice (Fig. 2k), we similarly calculated correlations to compare all pairs of trials from the two sessions.

**Neural population latent dynamics.** To estimate the latent dynamics associated with the recorded neural activity in each session for both mice and monkeys, we computed a smoothed firing rate as a function of time for each unit. We obtained these smoothed firing rates by applying a Gaussian kernel ($\sigma = 50$ ms) to the binned square-root transformed spike counts (bin size 30 ms) of each unit. We excluded units with a low mean firing rate (less than 1 Hz mean firing rate across all bins) but we did not perform any further exclusions, for example, based on lack of modulation or behavioural tuning. For each session, this produced a neural data matrix $X$ of dimension $n$ by $T$, where $n$ is the number of recorded units and $T$ the total number of time points from all concatenated trials on a given day; $T$ is thus given by the number of targets per day × number of trials per target × number of time points per trial. We performed this concatenation as described above after randomly subselecting the same number of trials for all targets for each animal (15 trials for monkey centre-out, six for monkey random walk, 22 for mouse reaching and pulling). For each session, the activity of $n$ recorded units was represented as a neural space—an $n$-dimensional sampling of the space defined by the activity of all neurons in that brain region. In this space, the joint recorded activity at each time bin is represented as a single point, the coordinates of which are determined by the firing rate of the corresponding units. Within this space, we estimated the low-dimensional latent dynamics by applying PCA to $X$. This yielded $n$ PCs, each a linear combination of the smoothed firing rates of all $n$ recorded units. These PCs are ranked on the basis of the amount of neural variance that they explain. We defined an $m$-dimensional, session-specific manifold by only keeping the leading $m$ PCs, which we referred to as neural modes. We chose a manifold dimensionality $m = 10$, based on previous studies examining motor cortical recordings during upper limb tasks[5,26,46]. Across all datasets, a ten-dimensional manifold explained about 60% of the neural variance for each of the monkey motor cortex (Extended Data Fig. 1c), mouse motor cortex and mouse striatum (Extended Data Fig. 5e). Note, however, that our

results held within a reasonable range of dimensionalities, similar to refs. 26,33,46 (Extended Data Figs. 2f and 4b). We computed the latent dynamics within the manifold by projecting the time-varying smoothed firing rates of the recorded neurons onto the $m = 10$ PCs that span the manifold. This produced a data matrix $L$ of dimensions $m$ by $T$.

**Aligning latent dynamics through CCA.** We addressed our hypothesis that different animals performing the same behaviour would share preserved latent dynamics by aligning the dynamics using CCA[26,75]. CCA was applied to the latent dynamics of each pair of sessions after concatenating the same number of randomly ordered trials to each target (condition, in the case of the sequential reaching task). For details on using CCA to align latent dynamics, see ref. 26.

We measured the similarity in latent dynamics across animals by computing the across-animal correlations as the canonical correlations (CCs) across all pairs of sessions from any two different monkeys or mice. To establish the strength of the across-animal correlations, we computed an upper bound defined by the within-animal correlations, which we calculated as the 99th percentile of the correlations between two randomly selected subsets of trials within any given session over 1,000 samples. The 'control' correlations represent a lower bound for the CCs. We computed these by shuffling the targets across the two sessions and using a randomly selected control window (more details in the 'control analyses' section below) in each trial, rather than the movement or preparatory epochs.

Note that to summarize each comparison to a single datapoint (for example, in Fig. 2k and Extended Data Figs. 2h and 6d), we computed the mean of the top four CCs of the latent dynamics[26]. In Fig. 2k, we used this approach to establish a relationship between the strength of preservation of the latent dynamics and the consistency of behaviour, quantified as the mean trajectory correlation of all possible pairs of trials across two animals. Furthermore, when showing pairs of 'aligned' trajectories across animals, such as in Fig. 2e and Extended Data Fig. 3, the CCA axes were made orthogonal using singular value decomposition for visualization purposes.

Finally, we showed that preserved latent dynamics could be uncovered across a broad range of manifold dimensionalities. In Extended Data Fig. 2f we repeated the alignment analysis for manifold dimensionalities $m = 2$–19.

**Decoding analysis.** To test whether the aligned latent dynamics maintain movement-related information, we built standard decoders to predict hand trajectory during overt behaviour. If the aligned latent dynamics across different animals were behaviourally relevant, they would allow predicting time-varying hand trajectories even if the methods used to identify them (PCA and CCA) are not supervised, that is, they do not attempt to optimize decoding performance. We compared the predictive accuracy of three different types of decoders: (1) a within-animal decoder trained and tested (using ten-fold cross-validation) on two non-overlapping subsets of trials from each session of each animal; (2) an across-animal 'aligned' decoder that was trained on the aligned dynamics from one animal and tested on another, a comparison we performed on each pair of sessions from two different animals; (3) an across-animal 'unaligned' decoder that was trained on the latent dynamics from one animal and tested on another without aligning the dynamics using CCA. We also performed a similar analysis to predict the upcoming target during covert movement preparation in monkeys (Fig. 4f).

Hand trajectory decoders were LSTM models with two LSTM layers, each with 300 hidden units, followed by a linear output layer. The models were implemented with Pytorch[76] and trained for ten epochs with the Adam optimizer, with a learning rate of 0.001. Upcoming target classifiers were Gaussian Naïve Bayes models[12] (the GaussianNB class in ref. 72). We included three bins of recent latent dynamics history, for a total of 90 ms, in the input of both the decoders and the

classifiers. These extra neural inputs incorporate information about intrinsic neural dynamics and account for transmission delays. The $R^2$ value, defined as the squared correlation coefficient between actual and predicted hand trajectories, was used to quantify decoder performance. Moreover, in Extended Data Fig. 4d we verified that our choice of across-animal decoder accuracy metric did not influence the observation that preserved latent dynamics are informative about behaviour by also computing a variance accounted for (VAF) metric, defined as:

$$\text{VAF} = 1 - \frac{\sum_{i=1}^{n}(\hat{y}_i - \bar{y})^2}{\sum_{i=1}^{n}(y_i - \bar{y})^2}$$

where $y_i$ represents the actual value of the predicted variable, $\hat{y}_i$ its predicted value and $\bar{y}$ its mean. For this analysis, we normalized hand trajectories by the length of the reaches (determined by the 99th percentile of their hand positions along each axis) because monkeys had workspaces of different sizes.

The hand trajectory was a two-dimensional signal in monkeys and a three-dimensional signal in mice. We built separate decoders to predict hand trajectories along the $x$, $y$ (and $z$ for mice) axes. We then reported the average performance across all axes. For target classification, we reported the mean accuracy of the classifier (the score() method).

To test how many dimensions of the aligned latent dynamics were needed for accurate across-animal decoding of behaviour, we repeated the decoding analysis in the monkey centre-out dataset for manifold dimensionalities $m = 1, 2…,14$ (Extended Data Fig. 4b).

Finally, we performed a control analysis to ensure our across-animal decoding results were not biased by sharing similar trials for both alignment and decoder training. We split the full dataset of one animal into three non-overlapping sets: one to align the latent dynamics, one to train the decoder and one to test the performance across animals. Extended Data Fig. 4c shows the result of this analysis for the monkey centre-out data. Despite having aligned the latent dynamics only using half of the data, the impact on decoding performance is negligible.

## Control analyses

**Alignment of latent dynamics with random behavioural windows.** To establish a 'behaviourally irrelevant' window as control, we randomly selected windows of similar length to our behavioural windows (450 ms) along the entire duration of the intertrial and trial periods combined. This ensured we had samples of dynamics in the neural population with realistic statistics but that they were not directly coupled to shared behaviour across individuals. We used this window to provide a lower-bound control for the alignment of neural population latent dynamics ('control' in Figs. 2f,g,j, 3d,e and 4b,e and Extended Data Figs. 2b–d,g, 3 and 8d).

**Aligning latent dynamics through Procrustes analysis.** We used CCA to align the latent dynamics in all the analyses. However, to ensure that our results hold regardless of the specific method used for alignment, we replicated the main result using Procrustes analysis[77]. Procrustes finds the best transformation that minimizes the sum of the squares of the differences between the two input datasets. Following a procedure identical to the CCA analysis, we aligned the dynamics from two different datasets using Procrustes analysis (the scipy.spatial.procrustes class in ref. 73) and then correlated the aligned dynamics to yield a metric comparable to that of the CCA (Extended Data Fig. 2g,h). Note that the degrees of preservation of latent dynamics obtained with CCA and Procrustes analysis are largely similar.

**Neural variance explained by aligned latent dynamics.** We measured the percentage of neural variance explained by the preserved latent dynamics using a method we devised in ref. 33. Briefly, we 'reconstructed' the preserved neural activity by projecting the aligned latent dynamics along the CC axes back to the PCA space (the neural

manifold) and then to the original neural state space. We then measured the difference between the total neural variance and the variance of these reconstructed signals using an approach similar to that in ref. 78. By repeating this procedure iteratively for an increasing number of manifold dimensions $m$, we measured the neural variance explained by each dimension of the aligned latent dynamics. Using this approach, we found that preserved latent dynamics explain a significant fraction of the neural population variance (Extended Data Fig. 2e).

**Surrogate datasets with TME.** We established a lower-bound control by aligning the latent dynamics from randomly selected windows sampled across different task conditions and behavioural epochs (see above). In addition to this control, we also used TME to generate surrogate neural data as another lower-bound control[29]. TME produces surrogate data that preserve the second-order statistics of the actual neural data (that is, covariance across time, across neurons or across experimental conditions) but are otherwise random (Extended Data Fig. 2a). Aligning these surrogate data through the same procedure as the original data shows significantly lower correlations for monkey centre-out task, random-walk task and mouse reaching and pulling task (Extended Data Fig. 2b–d).

**Aligning topological structure in neural population activity.** To test whether the topological structure in the produced movements is sufficient to produce preserved latent dynamics, we quantified the degree of similarity in latent dynamics across individuals that could be uncovered when aligning the static, topological features of the neural population activity, rather than the dynamics of the movements, using a technique developed in ref. 26. To align the topological structure of neural population activity, we time-averaged the activity for each neuron during the execution epoch of each trial in the monkey centre-out reaching task. We then analysed the time-averaged data with the previous methodology by performing PCA to find a neural manifold and using CCA to align each pair of sessions (Extended Data Fig. 7a). This procedure led to well-aligned 'topological representations' (example in Extended Data Fig. 7b). To directly test whether aligning the topological structure of neural population activity is sufficient to uncover preserved latent dynamics, we projected the latent dynamics on the CC axes found through this (static) topological alignment and calculated the pairwise correlations of the resultant projected latent dynamics. These correlations were remarkably lower than those obtained through alignment of the time-varying latent dynamics (Extended Data Fig. 7c,d).

**Control analyses on the numbers of conditions and neurons.** To establish that the preservation of latent dynamics holds across different degrees of task complexity, we calculated the correlations for increasing numbers of subsampled target conditions for each pair of sessions in the monkey random target task (Fig. 3f and Extended Data Fig. 8b). We randomly subsampled different combinations of target conditions and calculated the degree of preservation of the latent dynamics for up to 10,000 combinations for each number of conditions.

To establish that preserved latent dynamics can be uncovered regardless of the specific measured neurons, we also calculated the correlations for varying numbers of neurons in the random target task (Fig. 3g and Extended Data Fig. 8c). For each pair of sessions, we either randomly kept neurons (Fig. 3d) or randomly dropped neurons (Extended Data Fig. 8c) in increments of ten until we ran out of measured neurons for either session and repeated this process 50 times, calculating the degree of preservation at each step. For both analyses, we calculated the mean correlations for the top four CCs across all subsamples for each pair of sessions.

**Comparison of different but related tasks.** The central hypothesis of this study is that preserved latent dynamics are the basis for the generation of similar behaviour across individuals from the same species. Here, we sought to further support this hypothesis by showing that the latent dynamics produced by two individuals engaged in the same task are more similar than the latent dynamics produced by the same individual performing two different but related tasks. To this end, we compared our results to our previous study on the relationship of neural population activity underlying different but related wrist manipulation or reach-to-grasp tasks in monkeys[33] (Extended Data Fig. 9).

### Recurrent neural network models

**Model architecture.** To show that the preservation of latent dynamics across animals engaged in the same task is not a trivial consequence of similar behaviour, we trained RNNs to perform the same centre-out reaching task as the monkeys. These models were implemented using Pytorch[76]. Similar to previous studies simulating motor cortical dynamics during reaching[27,79–81], we implemented the dynamical system $\dot{\mathbf{x}} = F(\mathbf{x}, \mathbf{s})$ to describe the RNN dynamics:

$$\tau \dot{x}_i(t) = -x_i + \sum_{j=1}^{N} J_{ij} r_j(t) + \sum_{k=1}^{I} B_{ik} s_k(t) + b_i + \eta_i(t)$$

where $x_i$ is the hidden state of the $i$th unit and $r_i$ is the corresponding firing rate following tanh activation of $x_i$. All networks had $N = 300$ units and $I = 3$ inputs, a time constant $\tau = 0.05$ s and an integration time step $dt = 0.01$ s. The noise $\eta$ was randomly sampled from the Gaussian distribution $\mathcal{N}(0,0,2)$ for each time step. Each unit had an offset bias, $b_i$, which was initially set to zero. The initial states $x_{t=0}$ were sampled from the uniform random distribution $\mathcal{U}(-0.2,0.2)$. All networks were fully recurrently connected, with the recurrent weights $J$ initially sampled from the Gaussian distribution $\mathcal{N}\left(0, \frac{g}{\sqrt{N}}\right)$, where $g = 1.2$. The time-dependent inputs $\mathbf{s}$ fed into the network had input weights $B$ initially sampled from the uniform distribution $\mathcal{U}(-0.1,0.1)$. These inputs consisted of a one-dimensional fixation signal which started at 2 and went to 0 at the go cue and a target signal that remained at 0 until the visual cue was presented. The two-dimensional target signal ($2 \cos \theta^{\text{target}}$, $2 \sin \theta^{\text{target}}$) specified the reaching direction $\theta^{\text{target}}$ of the target.

The networks were trained to produce two-dimensional outputs $\mathbf{p}$ corresponding to $x$ and $y$ positions of the experimentally recorded reach trajectories, which were read-out via the linear mapping:

$$p_i(t) = \sum_{k=1}^{N} W_{ik} r_k(t)$$

where the output weights $W$ were sampled from the uniform distribution $\mathcal{U}(-0.1,0.1)$.

**Model training.** Networks were trained to generate positions of reach trajectories using the Adam optimizer[82] with a learning rate $l = 0.0005$, first moment estimates decay rate $\beta_1 = 0.9$, second moment estimates decay rate $\beta_2 = 0.999$ and $\epsilon = 1 \times 10^{-8}$. The loss function $L$ was defined as the MSE between the two-dimensional output and the target positions over each time step $t$, with the total number of time steps $T = 400$. The first 50 time steps were not included to allow network dynamics to relax:

$$L = \frac{1}{2B(T-50)} \sum_{b=1}^{B} \sum_{t=50}^{T} \sum_{d=1,2} (p_d^{\text{target}}(b, t) - p_d^{\text{output}}(b, t))^2.$$

To examine whether two networks could have different latent dynamics while producing the same motor output, we devised a network with more constraints to perform the behavioural task with distinct latent dynamics (Fig. 5a). We added a loss term that penalized the CC between the latent dynamics of the 'constrained' network being trained and those of another previously trained 'standard' network during movement execution:

$$L_{\text{constrained}} = L + \alpha \sum_{i=1}^{4} c_i^2$$

where $c_i$ is the $i$th CC. To examine different degrees of preserved latent dynamics, we trained the networks at varying values of $\alpha = 0$, 0.05, 0.25 or 0.50.

Networks were trained until the average loss of the last ten training trials fell below a threshold of 0.2 cm$^2$, for at least 50 and up to 500 training trials, with a batch size $B = 64$. Each batch had equal numbers of trials for each reach direction. We clipped the gradient norm at 0.2 before the optimization step. Both standard and constrained training were performed on ten different networks initialized from different random seeds. The same set of random seeds was used for constrained networks at different values of $\alpha$.

**Connectivity analyses.** By increasing the value of $\alpha$, we were able to decrease the preservation of the latent dynamics while keeping behavioural performance constant. To examine how this changed the underlying connectivity, we calculated the variance and dimensionality of the weight changes in the recurrent weights $J$ following training (Fig. 5f,g).

## Statistics and reproducibility

We compared the performance of various within-animal and across-animal movement decoders and classifiers using two-sided Wilcoxon's rank sum tests. We replicated the core findings across two species (mice and monkeys), four behaviours (a centre-out reaching task, a sequential reaching task and a reach, grasp and pull task, along with during covert movement planning) and two brain regions (motor cortex and dorsolateral striatum). Experiments on each species were performed independently in two different laboratories and by different scientists. The mice experiments were done in a single cohort, whereas the monkey data were collected in two sets of experiments (one for the centre-out task, another for the random reaching task), each spanning 2 years. Overall, our neural recordings and behavioural data are in good agreement with related published studies. All attempts at replication were successful.

## Reporting summary

Further information on research design is available in the Nature Portfolio Reporting Summary linked to this article.

## Data availability

Most of the monkey datasets are publicly available on Dryad (https://datadryad.org/stash/dataset/doi:10.5061/dryad.xd2547dkt) and CRCNS (https://doi.org/10.6080/K0FT8J72). The remaining monkey datasets and the mouse datasets will be made available on reasonable request.

## Code availability

All analyses were implemented using custom Python code and using open-source software. All the result panels are reproducible by running Jupyter notebooks. Code to reproduce all the results is openly available in https://github.com/BeNeuroLab/2022-preserved-dynamics.

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

**Acknowledgements** We thank K. Mitchell for suggesting the concept of the genome encoding a 'generative model' and other helpful comments regarding this research. We also thank L. Li for his contribution to the decoder analysis, F. H. Taschbach for suggesting further controls for the decoding analysis and C. Massumoto for the monkey and mouse illustrations. This work was supported in part by: grant no. H2020-MSCA-IF-2020-101025630 from the Commission of the European Union (M.S.), grant no. 108908/Z/15/Z from the Wellcome Trust (J.C.C.), grant nos NS053603 and NS074044 from the NIH National Institute of Neurological Disorders and Stroke (L.E.M.), grant (chercheurs-boursiers en intelligence artificielle) from the Fonds de recherche du Quebec Santé (M.G.P.), grant no. EP/T020970/1 from the UKRI Engineering and Physical Sciences Research Council (J.A.G.) and grant no. ERC-2020-StG-949660 from the European Research Council (J.A.G.).

**Author contributions** M.S., M.G.P. and J.A.G. devised the project. J.P. and J.T.D. provided the mouse datasets. M.G.P. and L.E.M. provided the monkey datasets. M.S. and J.C.C. analysed data and generated figures. J.C.C. trained and analysed the neural network models. M.S., J.C.C., J.T.D., M.G.P. and J.A.G. interpreted the data. M.S., J.C.C., M.G.P. and J.A.G. wrote the manuscript. All authors discussed and edited the manuscript. M.G.P. and J.A.G. jointly supervised the work.

**Competing interests** J.A.G. receives funding from Meta Platform Technologies, LLC. The remaining authors declare no competing interests.

**Additional information**
**Correspondence and requests for materials** should be addressed to Matthew G. Perich or Juan A. Gallego.

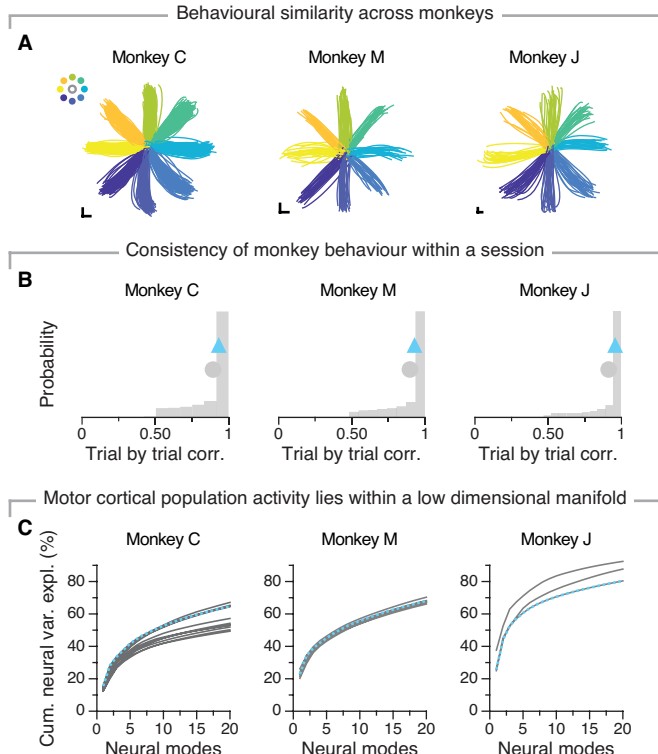

**Behavioural similarity across monkeys**

**A**

Monkey C          Monkey M          Monkey J

**Consistency of monkey behaviour within a session**

**B**

Monkey C          Monkey M          Monkey J

Probability

0    0.50    1          0    0.50    1          0    0.50    1
Trial by trial corr.      Trial by trial corr.      Trial by trial corr.

**Motor cortical population activity lies within a low dimensional manifold**

**C**

Monkey C          Monkey M          Monkey J

Cum. neural var. expl. (%)

0  5  10  15  20          0  5  10  15  20          0  5  10  15  20
Neural modes            Neural modes            Neural modes

**Extended Data Fig. 1 | Behavioural and neural data from monkeys performing a centre-out reaching task. A**) Hand trajectories of a single session of every monkey included in this work. Each line represents one trial, colour coded by target (inset). Scale bar, 1 cm. **B**) Distribution of the behavioural correlations for all pairs of trials from every included session for each of the three monkeys. Grey circle, mean. Blue triangle, mean for the representative session shown in (**A**). **C**) Cumulative neural variance explained as a function of the number of neural modes included. Data include 12 sessions for monkey C, 6 sessions for monkey M and 3 sessions for monkey J. Each line, one session. Dashed blue line, the same session shown in (**A**).

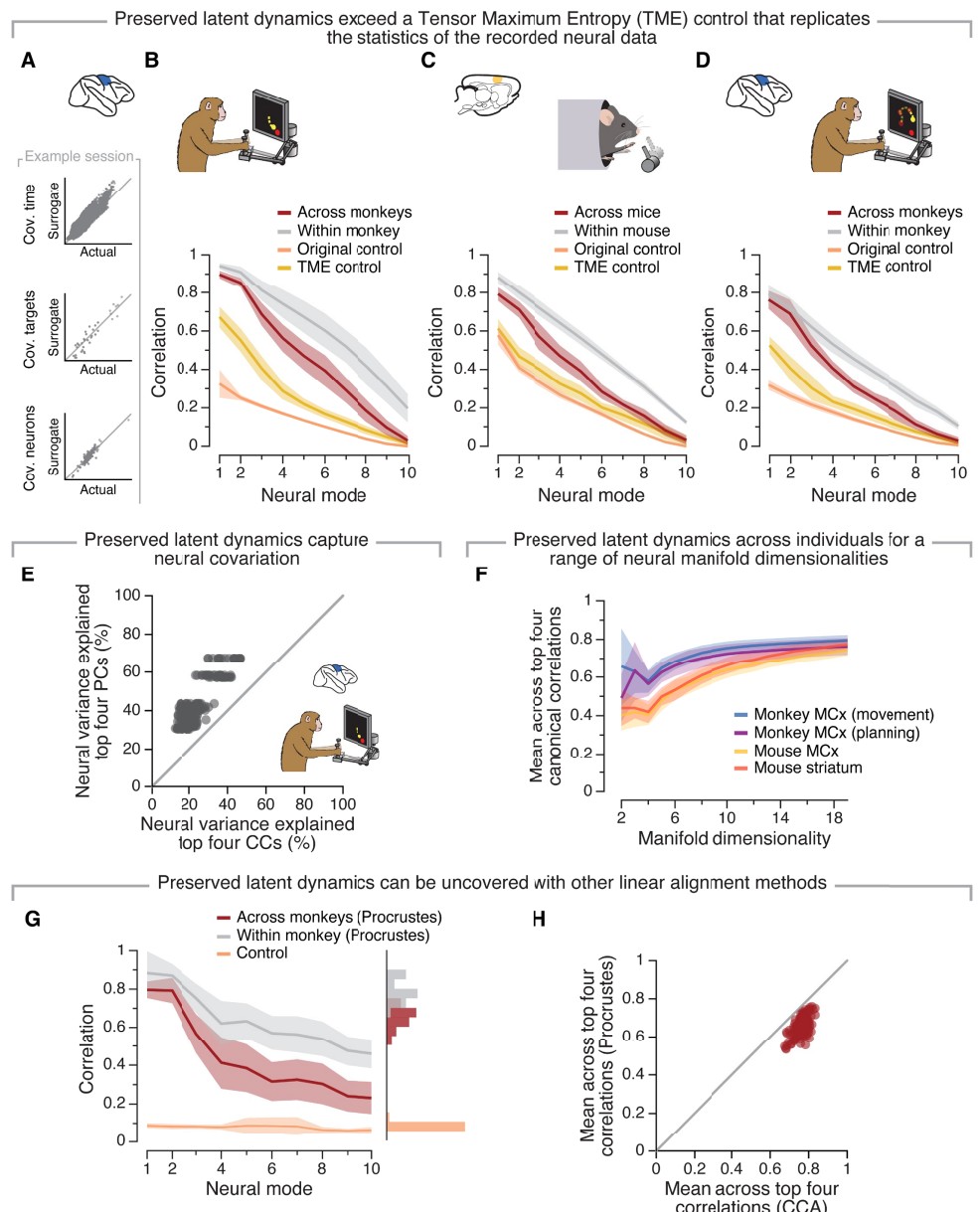

Preserved latent dynamics exceed a Tensor Maximum Entropy (TME) control that replicates the statistics of the recorded neural data

Preserved latent dynamics capture neural covariation

Preserved latent dynamics across individuals for a range of neural manifold dimensionalities

Preserved latent dynamics can be uncovered with other linear alignment methods

**Extended Data Fig. 2 | Additional verification of preserved latent dynamics. A**) We used Tensor Maximum Entropy (TME) to generate surrogate data that preserved all the second-order statistics of the actual neural data. Element-wise conservation of the covariance across time, targets and neurons between recorded and surrogate data for one example monkey centre-out dataset. **B**) Preserved latent dynamics across all pairs of sessions from three different monkeys performing the centre-out reaching task. Data include all 21 sessions and 126 comparisons shown in Fig. 2g. Note that the correlation between aligned latent dynamics across monkeys (red) was quite close to the within-monkey correlations (grey) and largely exceeded lower-bound controls from both randomly sampled behavioural epochs (orange) and TME (mustard). Line and shaded area, mean ± s.d. **C**) Similar to (**B**) but for mouse motor cortex during the reaching and pulling task. Data include all six sessions and 13 comparisons across four different mice shown in Fig. 2j. Line and shaded area, mean ± s.d. **D**) Similar to (**B**) but for the monkey motor cortex during the sequential reaching task. Data include all 11 sessions and ten comparisons across three different monkeys shown in Fig. 3e. Line and shaded area, mean ±

s.d. **E**) Preserved latent dynamics capture dominant features of neural population activity. Comparison of the total neural variance explained by the top four PCs (neural modes) and the top four Canonical Correlations. Single dots, individual comparisons for all monkey centre-out datasets. **F**) Preservation of latent dynamics does not depend on the number of manifold dimensions. Average canonical correlation as function of the dimensionality of the neural manifold (number of principal components) for four of the datasets considered (legend). Data pooled across all comparisons separately for each dataset. Line and shaded area, mean ± s.d. across the top four canonical correlations. MCx, motor cortex. Line and shaded area, mean ± s.d. **G**) Preserved latent dynamics can be uncovered with a different linear alignment method, Procrustes analysis (details in Methods). Data presented as in Fig. 2g but using Procrustes analysis rather than CCA to align the latent dynamics. Line and shaded area, mean ± s.d. **H**) Pairwise comparison between CCA and Procrustes analysis. Single dots, individual comparisons. Data includes 21 sessions and 126 comparisons across three monkeys.

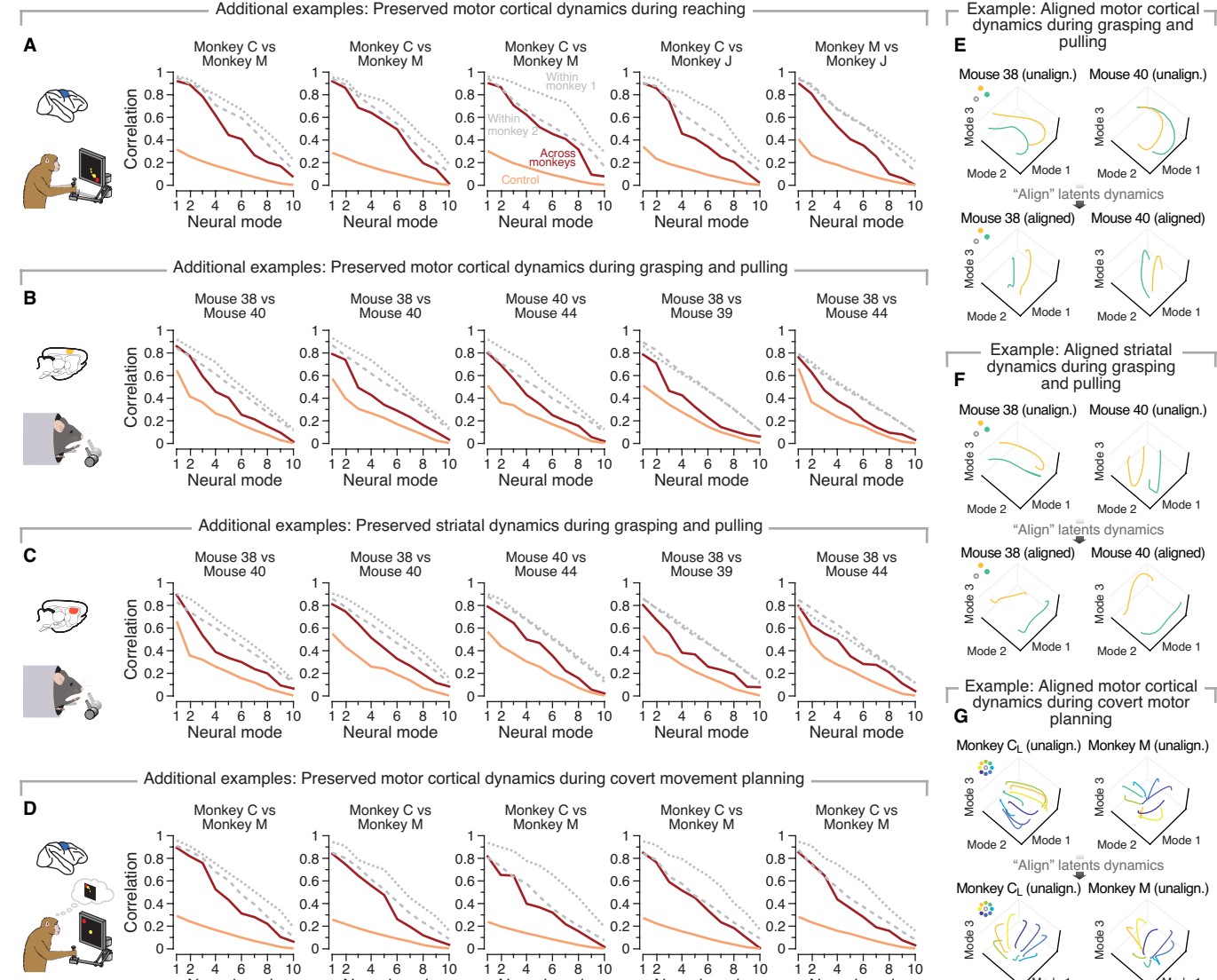

**Extended Data Fig. 3 | Additional examples of canonical correlations and preserved latent dynamics between pairs of sessions from two different monkeys and mice. A**) Example canonical correlations between motor cortical latent dynamics for monkeys during movement execution for the centre-out reaching task. Shown are five of the individual comparisons included in the pooled results in Fig. 2g. **B**) Example canonical correlations between motor cortical latent dynamics for mice during movement execution. Shown are five of the individual comparisons included in the pooled results in Fig. 2j. **C**) Same as (**B**) for mouse striatum. **D**) Same as (**A**) but during movement planning. **E**) Trajectories described by the motor cortical latent dynamics before (left) and after (right) alignment for a representative pair of sessions from two different mice. Individual lines, mean latent trajectory to each target (colour coded as in the inset). **F**) Similar to (**E**), for striatal latent dynamics in mice. **G**) Similar to (**E**), for motor cortical latent dynamics during covert movement preparation in monkeys.

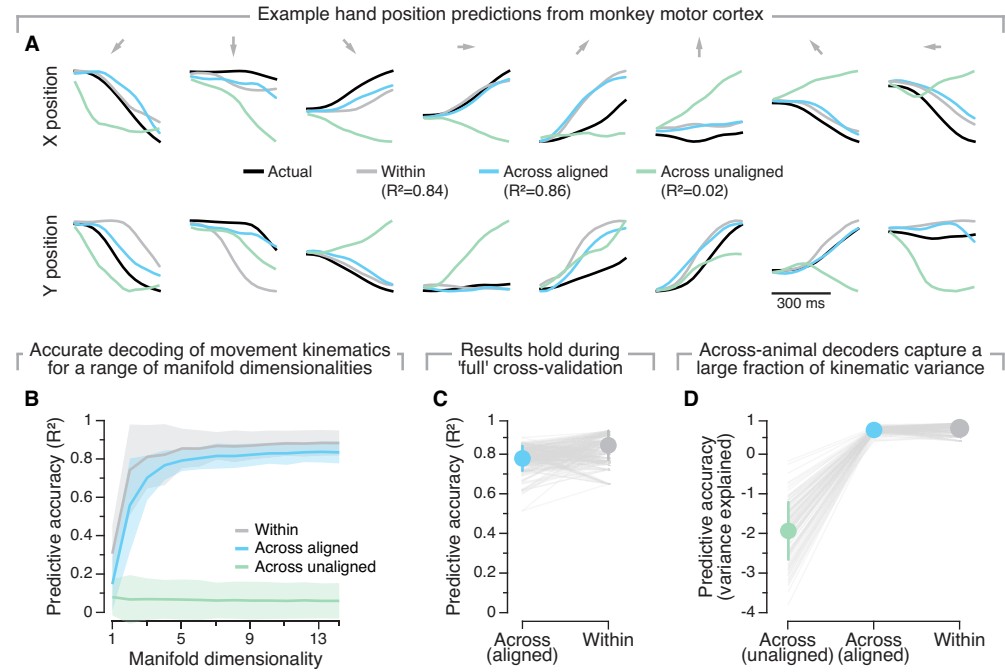

**Extended Data Fig. 4 | Preserved latent dynamics across monkeys are behaviourally relevant. A**) Example kinematic predictions across animals. Actual and decoded hand trajectory during one reach to each of the eight targets along the *X* (horizontal, top) and *Y* (vertical, bottom) directions; arrows on top of each column indicate target direction. The figure shows predictions using three models along with their respective accuracy (legend). Data from the same representative session shown in Fig. 2b. Note that a decoder trained on a different monkey after alignment of their latent dynamics ('Across aligned') performed virtually as well as a decoder trained and tested on the same session from the monkey being tested ('Within'). **B**) Decoding of hand position based on the latent dynamics for manifolds with increasing dimensionality. Line and shaded area, mean ± s.d. **C**) Across-animal decoding accuracy is not significantly impacted by a 'full' cross-validation procedure in which we used three sets of trials for training and testing (one for alignment, one for decoder training and one for testing). Error bars, mean ± s.d. **D**) Aligned latent dynamics also allow for accurate across-animal decoding of movement kinematics according to a variance explained metric (variance accounted for; Methods). Data points, individual comparisons between two sessions from different monkeys (*n* = 126). Error bars, mean ± s.d.

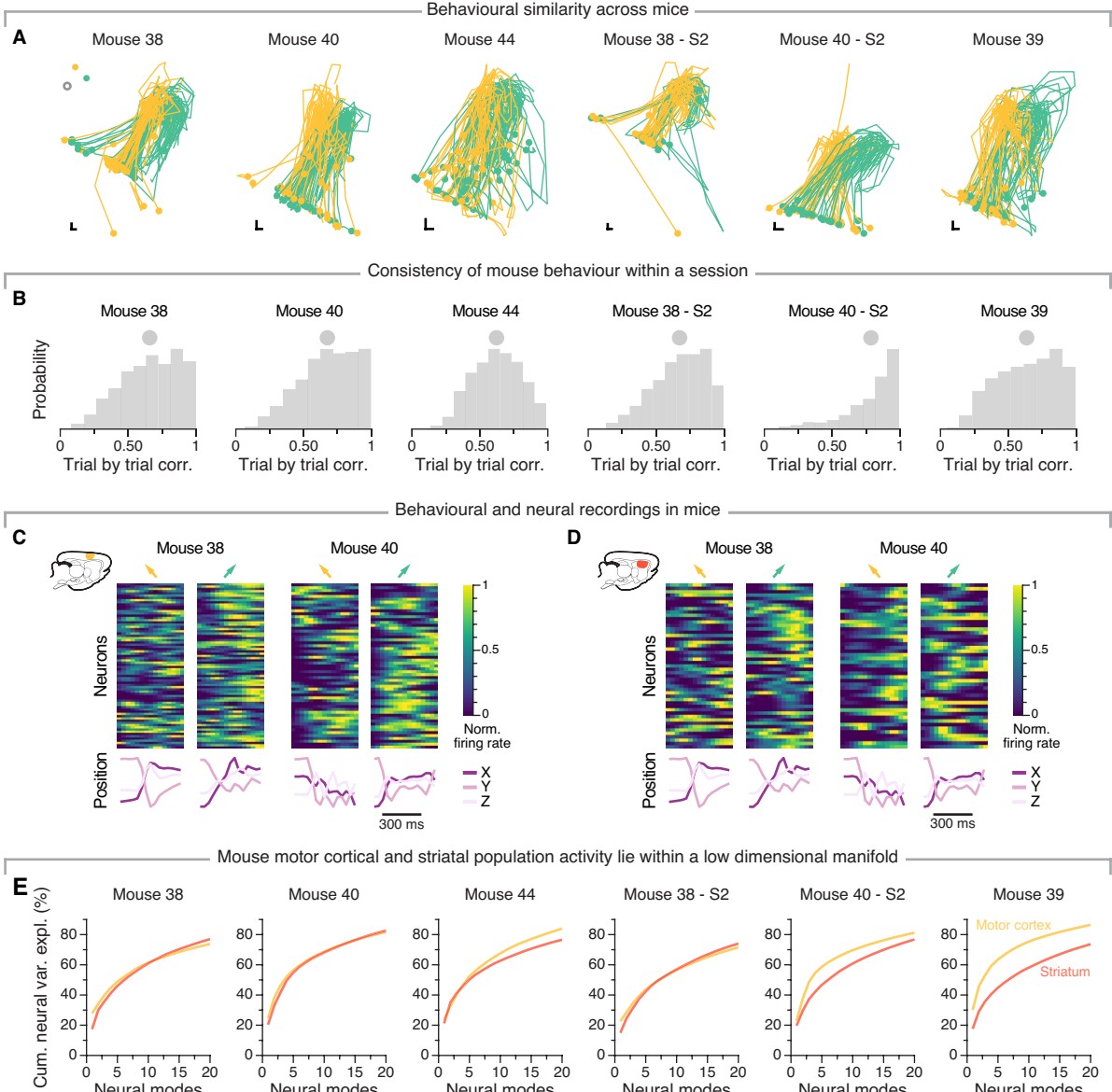

**Extended Data Fig. 5 | Behaviour and neural activity for mice performing a grasping and pulling task. A**) Hand trajectories of every session included in this work (notice that two mice have two sessions each). Each line represents one trial, colour coded by target (left, yellow; green, right; see inset). Circle, initial hand position. Scale bar, 1 mm. **B**) Distribution of the behavioural correlations for all pairs of trials from every session. Grey circle, mean. **C**) Example normalized motor cortical firing rates aligned to movement onset for two different mice (top) and corresponding hand trajectories (bottom). Each column, one reach to each of the two directions. Arrows on top of each column, reach direction, colour coded as in (**A**). **D**) Similar to (**C**) but showing the striatal activity during the same example movements. **E**) Cumulative neural variance explained as a function of the number of neural modes included, colour coded by brain region.

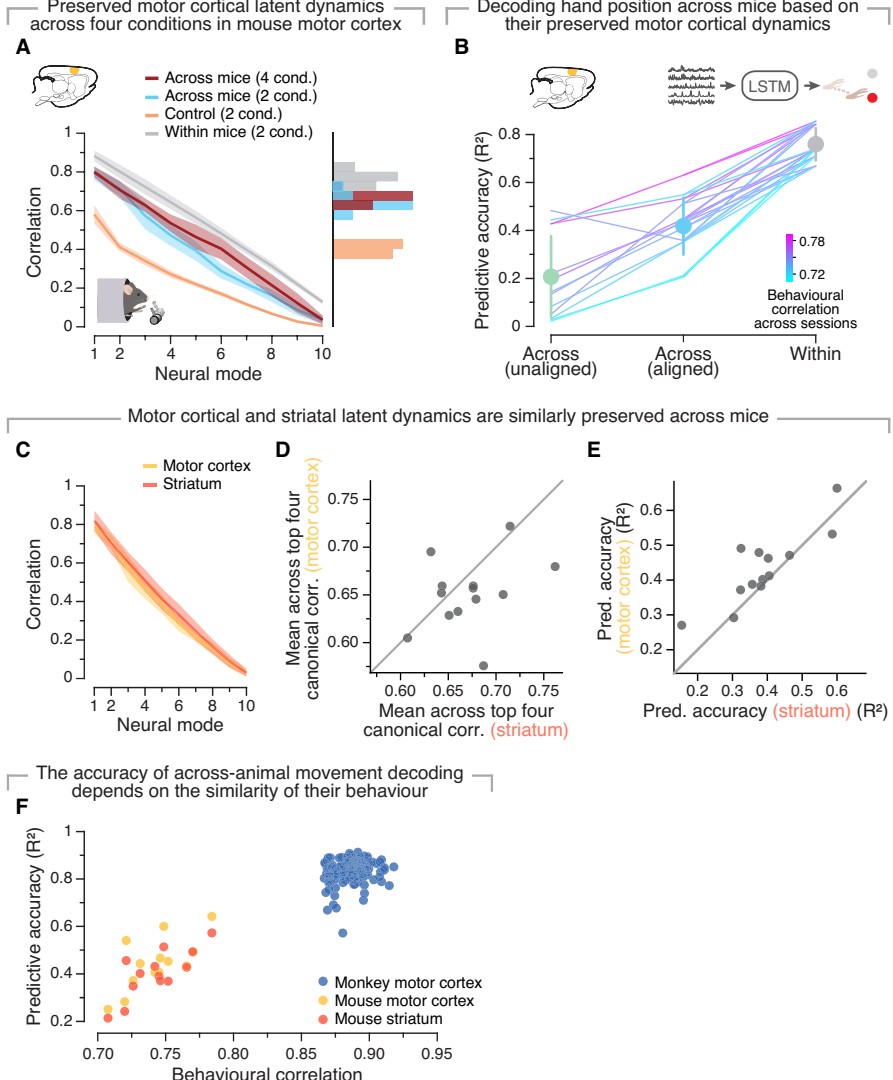

**Extended Data Fig. 6 | Further exploration of preserved latent dynamics in motor cortex and striatum. A)** Preservation of motor cortical latent dynamics during the four conditions (two reach directions × two loads) of the peri-grasp interval of the mouse grasping and pulling task (red) extends the results during the two reaching conditions (blue) from Fig. 2j. The lower-bound control (orange) and the within-animal upper bound (grey) from Fig. 2j are shown here for reference. Data include six sessions across four different mice ($n = 13$ comparisons). Line and shaded area, mean ± s.d. **B)** Decoders trained on the aligned latent dynamics from one mouse allowed predicting hand position from a different mouse with reasonable accuracy (blue), considerably better than decoders based on the unaligned latent dynamics (green). Lines, individual comparisons between two sessions from different mice ($n = 6$ sessions and $n = 13$ pairs), colour coded based on the correlation in hand kinematics across them (legend). Error bars, mean ± s.d. **C–E)** Motor cortical and striatal latent dynamics are similarly preserved during the mouse pulling and grasping task and allow for comparable decoding of movement kinematics. The across-animal motor cortical (yellow) and striatal (orange) correlations averaged across all comparisons (reproduced from Fig. 2j and Fig. 4b) are largely overlapping (**C**). Line and shaded area, mean ± s.d. Pairwise comparison of the degree of preservation of the latent dynamics (**D**) and across-animal decoder accuracy (**E**) for motor cortex and striatum. Single dots, pairs of sessions from two different mice ($n = 6$ sessions and $n = 13$ pairs). **F)** Decoding performance based on the aligned latent dynamics for each pair of sessions for the three stereotypical movement execution datasets (legend) as function of their mean behavioural correlation. Similar to Fig. 2k. Dots, individual comparisons across pairs of sessions that includes 21 sessions and 126 comparisons across three monkeys and six sessions and 13 comparisons across four mice.

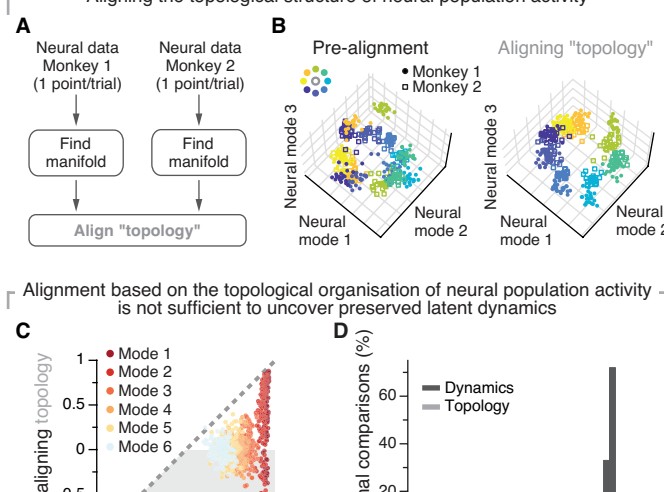

Aligning the topological structure of neural population activity

**A**

Neural data Monkey 1 (1 point/trial) → Find manifold → Align "topology"

Neural data Monkey 2 (1 point/trial) → Find manifold

**B**

Pre-alignment — Aligning "topology"

• Monkey 1  □ Monkey 2

Neural mode 3, Neural mode 1, Neural mode 2

Alignment based on the topological organisation of neural population activity is not sufficient to uncover preserved latent dynamics

**C**

Corr. aligning topology (vertical axis) vs Corr. aligning dynamics (horizontal axis)

Legend: Mode 1, Mode 2, Mode 3, Mode 4, Mode 5, Mode 6

**D**

Across animal comparisons (%) vs Mean across top four correlations

Dynamics, Topology

**Extended Data Fig. 7 | Aligning the topological organization of the neural data is not sufficient to uncover preserved latent dynamics across individuals. A**) To align the latent dynamics across sessions based on the topological organization of the neural data, we performed CCA on the population activity averaged across all time points on each trial. This process created a distribution of points with clear topological structure corresponding to the different reaching directions. **B**) This method provided excellent alignment of the topology on a single trial basis. Before alignment (left), the projection of each trial onto the neural manifold from each monkey (shown here overlaid on top of each other) looked different. After alignment, similar target-specific structure is present across both animals. Individual markers, a reach to one of the eight targets (colour code in inset) for Monkey $C_L$ (closed circles) and Monkey $M$ (open squares). **C**) Pairwise comparisons of the correlations after projecting the latent dynamics onto the CC axes found by aligning the topology (vertical axis) and onto the CC axes found by aligning the latent dynamics (horizontal axis). Data shown for the top six neural modes (see legend for colour code). Each dot represents one comparison; data include 21 sessions and 126 pairs of comparisons from three different monkeys during the centre-out reaching task. All dots lie well below the diagonal (dashed grey), indicating that aligning the latent dynamics based on the topology does not reach the correlation values obtained by aligning the latent dynamics. **D**) Correlation values were significantly lower when the alignment was based on preserving the topological structure of the neural population activity rather than its latent dynamics, illustrating the importance of the precise temporal dynamics for uncovering preserved latent dynamics. Same data as (**C**).

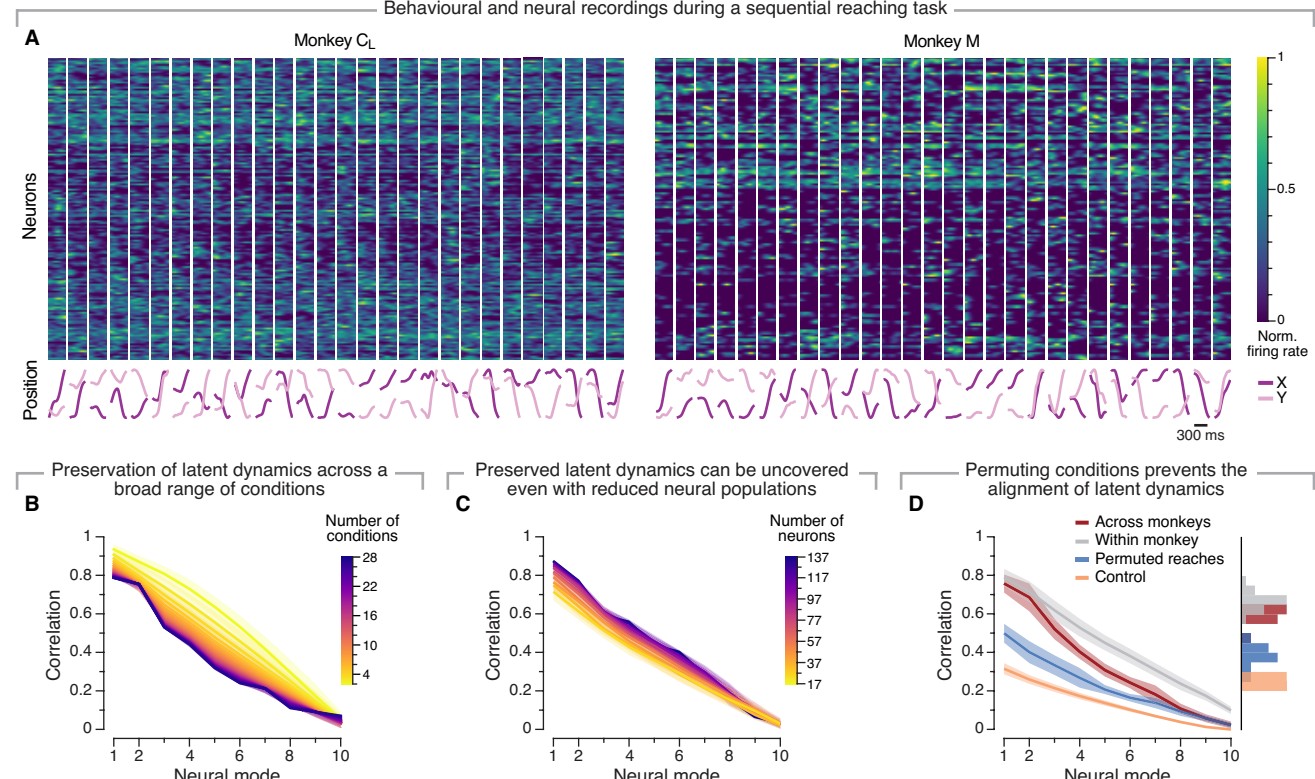

**Extended Data Fig. 8 | Behaviour, neural data and preservation of latent dynamics for monkeys performing a sequential reaching task. A**) Example normalized neural firing rates aligned to movement onset for two different monkeys (top) and corresponding hand trajectories (bottom). Each column, one reach to each of the 28 conditions, which we defined based on the starting location, duration and direction of the reaches. **B**) Canonical correlations as function of the number of conditions considered (legend). Note that canonical correlations stabilize after ~9 conditions are considered. Data from the same

example dataset as in (**A**). **C**) Canonical correlations as function of the number of neurons (legend). Note that the traces line up well after ~57 neurons are considered. Data from the same example dataset as in (**A**). **D**) Permuting the condition labels (blue) considerably decreases the alignment of latent dynamics across animals, bringing the canonical correlations closer to the lower-bound control. Data from all 11 sessions and ten comparisons across three different monkeys. Line and shaded area, mean ± s.d.

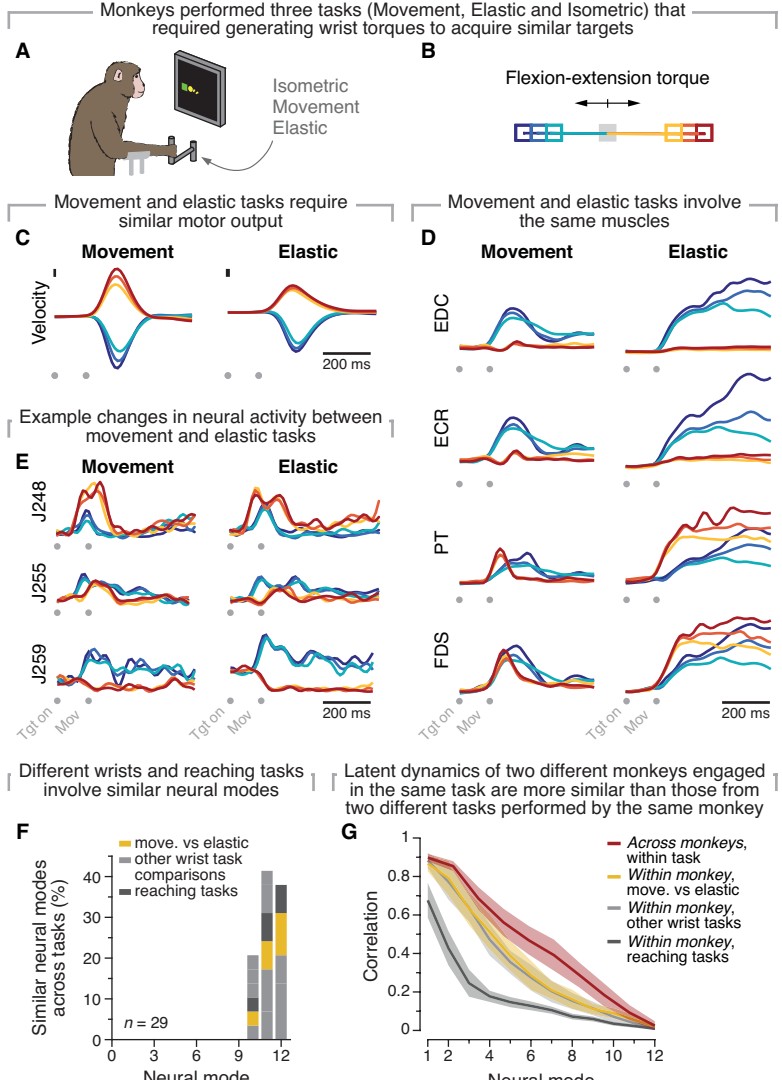

**Extended Data Fig. 9 | The latent dynamics of two different monkeys performing the same task are more preserved than those of the same monkey performing two related tasks. A**) We compared the latent dynamics produced by monkeys performing three related visually-guided one-dimensional wrist manipulation tasks using the same manipulandum. We also compared two reaching and grasping tasks (details in ref. 33). **B**) The organization of the targets and the produced cursor trajectories were the same across all three wrist manipulation tasks (note the overlapping cursor trajectories). **C**–**E**) Here, we show behavioural and neural data from the two wrist manipulation tasks that had the most similar kinematics ((**C**); lines colour coded by targets according to (**B**)): the one-dimensional movement task and the one-dimensional spring-loaded ('elastic') movement task. These two tasks engaged the same muscles (**D**) and the activity of single units was also relatively similar across them ((**E**) shows three example units). **F**) The covariance patterns elicited by these

tasks as well as the rest of wrist and reaching tasks was also quite similar across behaviours (this figure shows the number of neural modes out of twelve that are well-aligned across tasks according to a principal angle analysis). **G**) Despite the similarities in task characteristics (**B**), motor output (**C**), engaged muscles (**D**) and even single unit tuning (**E**), the latent dynamics generated by the same monkey as it performed different wrist tasks (mustard and light grey) or different reaching task (dark grey) within the same day were less similar than those of two different monkeys engaged in the same task (red). The across monkey results (red) illustrate preservation of the latent dynamics as monkeys performed the same centre-out reaching task; the results are the same as in Fig. 2e only recalculated with 12 components to match the across-task analysis in Gallego et al.[33] Lines and shaded area, mean ± s.d. across the comparisons indicated in the legend. (**A**-**F**) are adapted from ref. 33; the across-task alignment data in (**G**) partly reproduces Supplementary Fig. 7 therein.

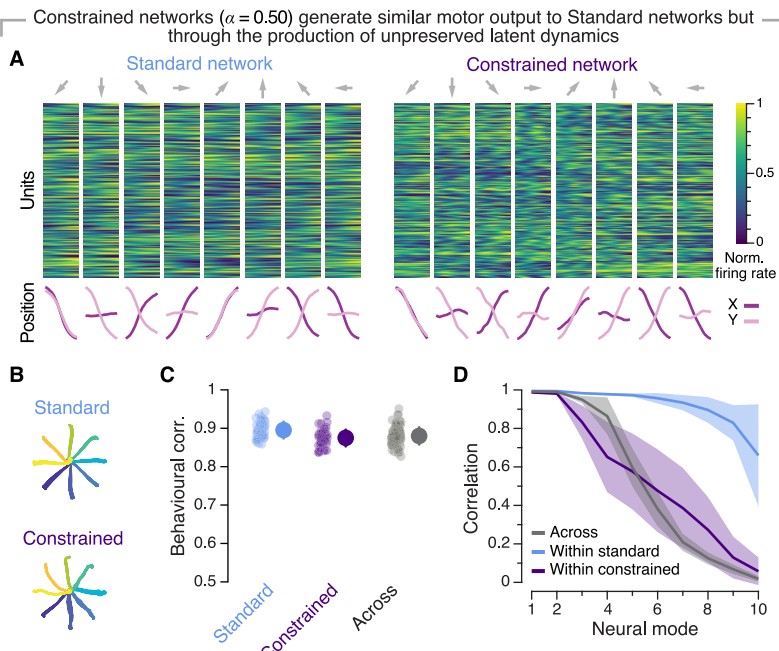

**Extended Data Fig. 10 | Additional data on recurrent neural network models. A**) Behavioural output and network dynamics for each reach direction for example Standard (left) and Constrained (right) networks that were penalized for canonical correlation similarity to the latent dynamics of Standard networks. Results presented as in Fig. 2b. Data for constrained networks represents results for $\alpha = 0.50$. **B**) Behavioural output produced by the two types of RNNs. The Standard (top) and Constrained (bottom) networks produce similar output. **C**) Behavioural correlation within and across pairs of Standard and Constrained networks. Data show 10 networks initialized from different random seeds for each type of network, 45 comparisons within network types, 100 comparisons across network types. Markers, individual comparisons. Error bars, mean ± s.d. **D**) Similarity in latent dynamics between pairs of Standard networks (blue) and Constrained networks (purple) and across network types (grey). Note that Constrained networks have correlations that are much more dissimilar than Standard networks. Likewise, the correlation between the latent dynamics of pairs Standard and Constrained networks are much lower than those between pairs of Standard networks. Thus, behavioural similarity is not sufficient to obtain highly correlated latent dynamics. Line and shaded area, mean ± s.d.

Matthew G. Perich

# Reporting Summary

## Statistics

For all statistical analyses, confirm that the following items are present in the figure legend, table legend, main text, or Methods section.

| n/a | Confirmed | |
|---|---|---|
| ☐ | ☒ | The exact sample size (*n*) for each experimental group/condition, given as a discrete number and unit of measurement |
| ☒ | ☐ | A statement on whether measurements were taken from distinct samples or whether the same sample was measured repeatedly |
| ☐ | ☒ | The statistical test(s) used AND whether they are one- or two-sided<br>*Only common tests should be described solely by name; describe more complex techniques in the Methods section.* |
| ☒ | ☐ | A description of all covariates tested |
| ☒ | ☐ | A description of any assumptions or corrections, such as tests of normality and adjustment for multiple comparisons |
| ☐ | ☒ | A full description of the statistical parameters including central tendency (e.g. means) or other basic estimates (e.g. regression coefficient) AND variation (e.g. standard deviation) or associated estimates of uncertainty (e.g. confidence intervals) |
| ☐ | ☒ | For null hypothesis testing, the test statistic (e.g. *F*, *t*, *r*) with confidence intervals, effect sizes, degrees of freedom and *P* value noted<br>*Give P values as exact values whenever suitable.* |
| ☒ | ☐ | For Bayesian analysis, information on the choice of priors and Markov chain Monte Carlo settings |
| ☒ | ☐ | For hierarchical and complex designs, identification of the appropriate level for tests and full reporting of outcomes |
| ☒ | ☐ | Estimates of effect sizes (e.g. Cohen's *d*, Pearson's *r*), indicating how they were calculated |

*Our web collection on statistics for biologists contains articles on many of the points above.*

## Software and code

Policy information about availability of computer code

| | |
|---|---|
| Data collection | Monkey dataset: Neural data was acquired using software from Blackrock Microsystems (Salt Lake City, USA); behavioural data was acquired using custom Matlab code (The Mathworks, Inc.). Mouse dataset: Neural data was acquired using the SpikeGLX software v. 3.0 (HHMI/Janelia Research Campus, Ashburn USA), preprocessed using Kilosort 2.0 (https://github.com/MouseLand/Kilosort), and manually curated using Phy (https://github.com/cortex-lab/phy}; behavioural data was estimated from videos using the video annotation tool JAABA (Kabra et al, Nature Methods, 2013); joystick operations were controlled using a custom Python package: https://github.com/janelia-pypi/mouse_joystick_interface_python. |
| Data analysis | All data analysis was performed in Python using open source packages such as numpy, matplotlib, sci-kit, scipy and pandas. All the analysis code is publicly available at https: //github.com/BeNeuroLab/2022-preserved-dynamics |

For manuscripts utilizing custom algorithms or software that are central to the research but not yet described in published literature, software must be made available to editors and reviewers. We strongly encourage code deposition in a community repository (e.g. GitHub). See the Nature Portfolio guidelines for submitting code & software for further information.

## Data

Policy information about availability of data

All manuscripts must include a data availability statement. This statement should provide the following information, where applicable:

- Accession codes, unique identifiers, or web links for publicly available datasets
- A description of any restrictions on data availability
- For clinical datasets or third party data, please ensure that the statement adheres to our policy

> The majority of the monkey datasets are publicly available on Dryad (https://datadryad.org/stash/dataset/doi:10.5061/dryad.xd2547dkt) and CRCNS (http://dx.doi.org/10.6080/K0FT8J72). The remaining monkey datasets and the mouse datasets will be made available upon reasonable request (these data have yet not been appropriately curated for public distribution).

## Research involving human participants, their data, or biological material

Policy information about studies with human participants or human data. See also policy information about sex, gender (identity/presentation), and sexual orientation and race, ethnicity and racism.

| | |
|---|---|
| Reporting on sex and gender | N/A |
| Reporting on race, ethnicity, or other socially relevant groupings | N/A |
| Population characteristics | N/A |
| Recruitment | N/A |
| Ethics oversight | N7A |

Note that full information on the approval of the study protocol must also be provided in the manuscript.

# Field-specific reporting

Please select the one below that is the best fit for your research. If you are not sure, read the appropriate sections before making your selection.

☒ Life sciences  ☐ Behavioural & social sciences  ☐ Ecological, evolutionary & environmental sciences

For a reference copy of the document with all sections, see nature.com/documents/nr-reporting-summary-flat.pdf

# Life sciences study design

All studies must disclose on these points even when the disclosure is negative.

| | |
|---|---|
| Sample size | We analyzed data from four monkeys (including two different behavioral tasks) and four mice to show reproducibility. Overall, our results include as many as 32 sessions across the two monkey datasets, and six sessions for the mice dataset. This sample size is larger than most studies in the field —in non-human primate research, a few sessions across n = 2 individuals is considered to be sufficient. In both species, we simultaneously recorded from enough neurons (or neural units in the case of one monkey), to reliably estimate the neural population latent dynamics that our results are based upon (see, e.g., Churchland et al Neuron 2010, Gallego et al Nature Comms, 2018, Sadtler et al Nature 2014, Trautman et al Neuron 2019). |
| Data exclusions | No data were excluded from analysis. |
| Replication | We replicated the core findings across two species (mice and non-human primates), four behaviors (a centre-out reaching task, a sequential reaching task, and a reach, grasp and pull task, along with during covert movement planning), and two brain regions (motor cortex and dorsolateral striatum). Experiments on each species were performed independently in two different laboratories, and by different scientists. The mice experiments were done in a single cohort, whereas the monkey data was collected in two sets of experiments (one for the centre-out task, another for the random reaching task), each spanning two years. Overall, our neural recordings and behavioural data are in good agreement with related published studies. All attempts at replication were successful. |
| Randomization | Covariates were not experimentally controlled so randomization was not necessary for our study (because all the animals were assigned to the same group). |
| Blinding | Blinding  was not necessary for our study as we were not assessing the effect of any manipulation or comparing across conditions. |

# Reporting for specific materials, systems and methods

We require information from authors about some types of materials, experimental systems and methods used in many studies. Here, indicate whether each material, system or method listed is relevant to your study. If you are not sure if a list item applies to your research, read the appropriate section before selecting a response.

## Materials & experimental systems

| n/a | Involved in the study |
|-----|----------------------|
| ☒ | ☐ Antibodies |
| ☒ | ☐ Eukaryotic cell lines |
| ☒ | ☐ Palaeontology and archaeology |
| ☐ | ☒ Animals and other organisms |
| ☒ | ☐ Clinical data |
| ☒ | ☐ Dual use research of concern |
| ☒ | ☐ Plants |

## Methods

| n/a | Involved in the study |
|-----|----------------------|
| ☒ | ☐ ChIP-seq |
| ☒ | ☐ Flow cytometry |
| ☒ | ☐ MRI-based neuroimaging |

## Animals and other research organisms

Policy information about studies involving animals; ARRIVE guidelines recommended for reporting animal research, and Sex and Gender in Research

| | |
|---|---|
| Laboratory animals | We recorded from four non-human primates (three macaca mulatta, one macaca fascicularis; all male, aged 6-10 years) and four mice (C57BL/6; aged 8-16 weeks). Mice were maintained on a 12-12 h (8am-8pm) light/dark cycle and recordings were made between 9am and 3pm. The holding room temperature was maintained at 21+/-1°C with a relative humidity of 30% to 70%. |
| Wild animals | No wild animals were used in this study. |
| Reporting on sex | Sex was not considered in this study. |
| Field-collected samples | This study did not involve samples collected from the field. |
| Ethics oversight | All surgical and experimental procedures for the monkey dataset were approved by the Institutional Animal Care and Use Committee of Northwestern University under protocol \#IS00000367. All surgical and experimental procedures for the mouse dataset were approved by the Institutional Animal Care and Use Committee of Janelia Research Campus. |

Note that full information on the approval of the study protocol must also be provided in the manuscript.

