## [Peer Review File · Nature]

Manuscript Title: Preserved neural dynamics across animals performing similar behaviour

Reviewer Comments & Author Rebuttals

Reviewer Reports on the Initial Version:

Referees' comments:

Referee #1 (Remarks to the Author):

Safaie et al., address the representational similarity of a neural manifold across individuals in two different animal species (mice and monkeys) and two different brain regions during a reaching task. They find interesting commonalities between individuals that can be harnessed for decoding performance. A decoder that is trained on one individual can decode the arm position in another individual – after an alignment procedure. Overall, the study by Safaie et al., represents a significant advancement in our understanding of the connection between neural latent dynamics implementing behaviour across individuals. The authors present convincing evidence on the existence of common latent dynamics among individuals performing the same task and present modelling results arguing that this similarity is not necessarily due to behaviour per se. This is important because they suggest that behavioural convergence can emerge from common underlying latent dynamics in the dynamics of anatomically different circuits. It is notable that the authors have arrived at their conclusions using only linear methods like CCA with samples from different models and regions. I am enthusiastic about the study and support its publication. However, two issues and several minor clarifications need to be addressed to better understand their findings and further strengthen the manuscript.

1. Latent dynamics and circuitry considerations

The authors could comment and analysis how latent dynamics relates to circuitry details. As the authors mention, low-level details of the circuitry do not necessarily account for behavioural similarities, however, for common latent dynamics to emerge there must be some structural similarities underpinning it – e.g., connectivity rank in the case of seminal work by Ostojic et al. The authors address this by training two RNNs to have dissimilar latent dynamics. However, it would be nice to conduct cross checks with controlled levels of latent structure dissimilarity (0, 30, 50 or 90%) to see how similar or dissimilar would the circuitry be then. Operating within their RNN model, one could modify the alpha parameter from equation 4 to see how it affects the underlying connectivity – i.e., different penalties on the CCA correlations.

Alternatively, authors could comment on how the similarity in the latent structure of motor cortex compares to the variability across individuals in, for example, their connectome (see Individual variability in functional connectivity architecture of the mouse brain, *Communications Biology*, 2020) or cell-type (see Phenotypic variation of transcriptomic cell types in mouse motor cortex, *Nature*, 2020). This could further improve the argument on how neural modes correlation is high in spite of variability in the underlying circuitry. Some quantification along those lines could be useful to

reinforce the message about circuitry consequences.

2. Analysis of the error trials in planning or covert behaviour

In both covert and overt behaviour, the planned and actual motion corresponds to a limited action space that has as its downstream output a concrete number of motor neurons. Hence, for a successful trial in the reaching task, future action plans must somehow converge even if the behaviour is a covert process. Is there any evidence of representations diverging if the intended movement does not coincide with the executed one? In other words, whether error trials can be decoded and, in such case, if the motor representations between individuals are correlated. This could constitute more convincing evidence of how planning is shared, and it is due to common latent dynamics.

In case error trials are not readily available, some brief discussion would help regarding the embodiment of the problem – i.e., how activity convergence may be due to the recruitment of the same motor neurons – or in what regions representations may be more divergent – e.g., state or choice maps.

3. More discussion on why striatum and motor cortex are dissimilar

The authors studied the latent dynamics both in the dorsolateral striatum of mice and M1 or PMd of monkeys and found that CCA improves predictive accuracy (Fig. 3) but much less effectively for striatum compared with M1 (Fig. 3 C,F). I couldn't locate information about whether the blue and the green data points for mice are significantly different, could the authors add the corresponding stars for all points? Additionally, there seems to be significant decoding drop relative to the M1 data from mice after alignment, could the authors try to unpack the origin of this? Is there less a less common latent space? Decoding within seems to show a high R^2 (0.6) but something fails for the CCA alignment in the striatum? How similar are trajectories (a visual would be helpful in the supplement)? Does the region code for something that is not strictly necessary for the same grasping motion and therefore is different across individuals? Does not align well across animals but aligns within the same animal on different trials? Is it a failure of the transfer to a different animal or poor alignment quality that could be better trial-to-trial within the same animal, a trial-vs-animal comparison would be helpful. Is there may be some additional normalisation needed, for speed or starting point to control for mouse-to-mouse behavioural differences? In any case it is interesting to comment more on why one would expect similar or dissimilar latent dynamics in the striatum compared to M1, its pathway position etc.

Minor suggestions and corrections:

1) Predictive accuracy quantified through correlations (R^2), which is high if two variables evolve conjunctly in their dynamics but does not account for errors in distance between the predicted and true variables (L-norm). Supplementary Figure S5 shows traces are indeed similar in the primate case, one similar could be present for mice or could be shown by a RMSE plots. LSTM decoders may be fitting well the temporal dynamics, but the putative positions may have more errors in distance. How does the prediction fare in terms of distance metrics?

2) There are cases (Figure 2H or S4) where the across prediction is better than the within ones. In

which cases the canonical space is benefitting the decoder?

3) Figure S3 shows that correlations are, to some extent, invariant to the manifold dimensionality. But do these results hold for low-rank cases (1-7 PCs)?

4) Correlations are shown by neural modes (Figure 2G, J and 3B-E), however, figures S1C and S5C present some diversity in the dimensionality of each individual by the number of components needed for a similar cumulative variance. Since $n=10$ is used for all post-CCA analysis, a concern is whether this divergence of dimensionality affects the analysis. Internal dynamics may be shared among individuals and be highly correlated or explanative of the variance but uninformative in terms of the behavioural readout. Plotting how decoding accuracy changes with a lower number of PCs could show this.

5) Errata: Figure S8 reads "similar to Figure 2I" where it should say Figure 2K instead.

Referee #2 (Remarks to the Author):

This paper provides an analysis of neural population recordings from monkey and mouse brains which shows that the brains of different animals of the same species use similar dynamics to perform the same task or behavior. This is a remarkable finding given that the brains of individual animals have different wiring diagrams, cell type densities, and that only a small subset of neurons in each brain or brain region could be recorded. (The converse was not true: the dynamics of monkey brains while performing a reaching task did NOT resemble the dynamics of mice performing a grabbing-and-pulling task). The authors used PCA to identify the neural manifold (low-D subspace of neural activity) in each recording, then used canonical correlation analysis (CCA) to align the latent dynamics from different brains. Moreover, they show that a decoder trained on the aligned neural data from one brain can be used to decode behavior from the aligned neural data of another brain. The paper is well written and likely to be of interest to a broad readership.

My biggest concern is about how surprising it is that CCA turns up similar dynamics from different individual brains. The paper compares the observed similarity with a null distribution based on shuffled data. (The Methods section is a bit sparse and I admit I did not fully understand the shuffle control, which refers to using a correlations computed using a "randomly selected control window").

However, I think it would be worthwhile to understand whether the level of correlation observed after CCA is something that would arise from any set of neural activities with the same temporal correlations and neuronal correlations. The most elegant (in my view) treatment of this issue came from a paper from Gamal Elsayed and John Cunningham:

Elsayed, G. F. & Cunningham, J. P. Structure in neural population recordings: an expected byproduct of simpler phenomena? *Nature Neuroscience*, 2017

In that paper, the authors set out to test whether the "rotational dynamics" reported in Churchland et al 2012 [ref 42] were truly surprising, or if they could be captured by any population that had the same temporal correlations, neuron-neuron (pairwise) correlations, and neuron-task correlations. They came up with a maximum-entropy model for generating new datasets that preserve those sets of pairwise statistics, and which could be used for testing whether the neural data from Churchland et al had rotational dynamics above and beyond that captured by the maximum-entropy null model.

Elsayad and Cunningham have released the code to fit their max-ent distributions to neural data, and so it seems like performing a similar test is something the authors of this paper could do without an excessive amount of additional work. If they find that the alignment of the neural data across monkeys is greater than that achieved by the max-ent distributions preserving the temporal, pairwise, and task correlations, then I will be substantially more convinced that the result is not a trivial consequence of lower-order statistical structure in the data. (Actually, I am undecided about the necessity of including the neuron-task correlations in the null model; certainly it is a stronger result if the neural data also exceed the alignment of this null distribution, but at least I would think it should at least be more significantly aligned than the temporal and neuron-neuron correlations null model).

Another idea that occurred to me while reviewing the paper: what if the authors sought to align the data across monkeys with a shift in the labeling of the targets? That is, use CCA to align the datasets so that the neural data from reaches to target 1 in monkey A are aligned with neural data from reaches to target 4 with in monkey B. And then align target 2 data in A with target 5 data in B, target 3 with 6, 4 with 7, etc. Are the datasets still as aligned if we are allowed to freely rotate the targets in one monkey compared to the other? This also would suggest that the alignment in the dynamics is unique to the specific behaviors, but also holds across shifts or permutations of those behaviors, which is a somewhat weaker result.

Or, what about the alignment between cortical and striatal data? Are the dynamics more similar within brain regions than across different brain regions? The results will again be more surprising if we find that the motor cortical dynamics of one animal are more similar to the motor cortical dynamics of a second animal than they are to striatal dynamics in either animal. (Although if not this would also tell us something useful).

Other comments:

1. CCA returns a set of axes ordered by the strength of correlation, and these (descending) correlations are plotted, for example, in Fig 2F,G, and J. However, what seems to be hidden here is how much variance in the original manifold these CCA axes account for. That is, to what degree are the dominant correlated directions aligned with the major principal components of the data? If the 3 most correlated axes of the aligned data are loading mostly on the bottom 3 Principal Components, while the bottom CCA axes have high correlation with the dominant PCs, then this tells us that the alignment is not *that* strong --- i.e., the alignment only holds if we squint and look at the principal component axes with smallest variance. I suspect that is not the case, but it would be nice to quantify this alignment one way or the other. (I'm not sure exactly what metric to use, but at the very least we could look at what the PC projection of the top CCA axes is, or try to quantify how

much variance is captured by the top CCA axes compared to the top PC axes). I apologize if the authors addressed this and I simply missed it!

2. One additional methodological concern I had was about the decoding of aligned data. To avoid double-dipping, the authors should compute the alignment from a training set, fit the decoder on this training set (or another training set, if desired), and then perform decoding on a held-out test set that was not used for computing the alignment (or fitting the decoder). From the methods section, it was unclear to me whether this was what was done or not.

3. A more general comment is that I found the Methods to be quite sparse in general. There are a number of results that are described as "cross-validated", but it is unclear exactly what this means -- i.e., how the train-test splits were chosen, and exactly what was done in each split. Likewise, I found it hard to understand what some of the controls were, or how the null distribution was constructed in some cases. I would suggest a careful rewriting of this section with a lot more detail (i.e., including things that may seem obvious to the authors, but will not be obvious to all readers!)

4. Another paper worth citing is this one from Kenneth Latimer:

Low-dimensional encoding of decisions in parietal cortex reflects long-term training history
Kenneth W. Latimer, David J. Freedman
doi: <https://doi.org/10.1101/2021.10.07.463576>

This paper reports a finding of different dynamics in the same task if animals have a different training history. This does not directly contradict the results in this paper (but it is suggestive that the similarity found here *might* rely on the similarity of training histories). It seems worth citing, at least!

5. Pg 7, line 115: "higher-order animals". I'm not sure it's still common to refer to species as higher and lower order; it might be worth finding a different adjective!

6. pg. 15:

297 The R² value, defined
298 as the squared correlation coefficient between actual and predicted hand trajectories, was used
to quantify
299 decoder performance.

I don't believe it is standard to report R² as the squared correlation coefficient, which can produce artificially rosy results (since the correlation coefficient is insensitive to errors in the mean and scale of the prediction). I prefer --- and I believe it is standard practice as well --- to report R² as the coefficient of determination, defined as $1 - SSR / SST$, where $SSR = \text{Sum of squared residuals}$, i.e., $1/N \sum (y_i - \hat{y}_i)^2$ and $SST = \text{sum of total squares}$, i.e. $1/N \sum (y_i - \text{mean}(y))^2$. (For more see https://en.wikipedia.org/wiki/Coefficient_of_determination).

Referee #3 (Remarks to the Author):

This study by Safaie et al examines an interesting and important issue of systems neuroscience, namely whether some invariance —although it is not described as such here— across time within individuals, and across individuals, can be detected in the activity of circuits involved in the planning and execution of behavior. The approaches center on the extraction and comparison of “latent dynamics” from multi-neuronal recordings taken within and across individuals in two sets of data/experiments (each in one species: NHPs and mice). The conclusions are that such similar brain (cortical and subcortical) latent dynamics appear to be detectable in and across individuals for several sets of movements.

Whereas the topic is really interesting, I was not convinced by the study (premise, data, and analyses presented in the manuscript) as it stands.

Premise and hypotheses.

“Preserved”. The point is minor but confusing: the authors use this term throughout including the title when I think they might mean “similar” or “invariant”. Preserved (conserved would be equivalent) implies (to me at least) a common origin or common source (developmental, evolutionary etc). If it is what was meant, then I failed to see the justification.

Lines 5, 25, 34. “Idiosyncratic” neural circuitry.

Line 23: “large differences in brain circuitry”

The premise of the study seems to be the assumption that the brains of different individuals are entirely different from one another. But we know this not to be true, at some degree of organization and description: cell types are many but reliably identifiable across individuals in a species; with the exception maybe of central olfactory circuits (and even there, evidence for lack of organization is not always convincing), maps, mappings and connectivity conform to known and predictable patterns (eg, within and across layers, within and across areas etc) with a great degree of graph precision (eg, per cell type); they are, like neuronal identities, the results of precise developmental programs, followed by local instructions and refinement via plasticity. Circuits are therefore far from being idiosyncratic —except if one thinks of circuits as composed of identifiable individual neurons, which they are probably not in NHPs and rodents; the real issue here is therefore one of resolution and granularity of the description: at an appropriate scale, circuit designs are not idiosyncratic.

To some extent, the fact that one can find similar rules of organization in say, visual or motor cortex from one individual to another, using massive under-sampling of those circuits (typically a tiny fraction of the neurons present), already indicates widespread regularities and invariances that one could expect to find also in low-dimensional projections. So the real problem, it seems to me, is one of resolution (of recorded data, and of output reconstruction and prediction). I imagine for example that if the primate motor task in the present ms. consisted in distinguishing 100 or even 50, rather than 8 targets, accuracy would drop dramatically, neural data would need to be increased equally dramatically, and alignments of datasets might not be very good. When in line 34-35, the authors say “...low level details of neural circuits ... should not be necessary to account for the emergence of

species-typical behaviors”, I wonder what is meant. Everything is a question of degree and of precision. My ability to play the violin and the kinematics of our movements likely differ from Itzhak Perlmann’s by many low-level details of our respective neural circuits. At some level of description, we must diverge, but there may be low-d approximations where those differences cannot be detected. It is therefore a bit circular.

In addition, because it is already known that similarities can be found across recordings made during motor execution and during the preparation for intended movement in motor/premotor areas, the fact that invariances may be found for movement execution would seem to lead logically to the prediction that preparation for action should similarly see invariances across trials and across individuals. The opposite finding would have been interesting, because unexpected (I think).

Methods.

I found it hard to put together many of the methodological details that matter in the study.

To start and most trivially, how many neurons were recorded in each species and experiment? Where they sorted single units? Multi-units?

What datasets were from already published experiments, what were new ones?

Data were reduced to 10D by PCA, and this choice of 10 modes is not precisely justified, or the consequences of this choice explored (except around 10 ± 3 in one analysis). These 10PCs, one finds in the methods, explain about 60% of the data. Is that a reasonable representativeness to address the questions posed? What does the spectrum of eigenvalues look like? Said differently, one would like to know at what degree of dimensionality reduction the equivalences or invariances (eg, between responses recorded from different individuals) stop being identifiable. Some of the plots (eg in Fig 2F and G) suggest that the dimensionality needs to be considerably reduced.

Inter-individual reduced representations were aligned by CCA. But if the representation of angle (in the NHP pointing task) in neural space is topologically ordered within individual animals, alignment by CCA across animal representations should not come as a surprise, or should it?

In the example with mouse data, the behavior is described as consisting of 4 tasks (two positions of a lever times two resistance levels), but the resistance level data for each position are merged in the analysis, making it in fact a 2-option task. It seems evident that aligning them across animals should not be difficult (the opposite would have been surprising). Hence the mouse data seem to provide very little of interest here.

Controls.

I found it very hard to find the information concerning the controls (eg fig 2F, G etc). On line 58, the authors state a control with “randomly selected behavioral epochs”. In the methods’ Data Analysis section (line 228-231), however, the control windows are called “behaviorally irrelevant” windows, and defined as 450ms segments “taken randomly along inter-trial and trial periods combined”. What does that mean exactly? Why are they called behavioral epochs in the results and are not so in the methods? How are these windows constructed/chosen? Those details would seem to matter a great

deal.

Test in lines 85-89: In this comparison, the authors report that the latent dynamics across animals for the same task are more similar than within the same animal across tasks. Again, the devil is in the detail of the movements and behaviors. If the movements are completely different, engaging different muscles, joints and forces, would this not be expected? And would it not be more appropriate to carry out CCA on (reduced) within-animal datasets recorded for the two behaviors, and then test across both animals and behaviors?

In short, while I found the topic of the study extremely interesting, I was not convinced by the data, analyses and interpretations as they stand now.

Response to reviewers

“Preserved neural population dynamics across animals performing similar behaviour”

Referee #1 (Remarks to the Author):

Safaie et al., address the representational similarity of a neural manifold across individuals in two different animal species (mice and monkeys) and two different brain regions during a reaching task. They find interesting commonalities between individuals that can be harnessed for decoding performance. A decoder that is trained on one individual can decode the arm position in another individual – after an alignment procedure. Overall, the study by Safaie et al., represents a significant advancement in our understanding of the connection between neural latent dynamics implementing behaviour across individuals. The authors present convincing evidence on the existence of common latent dynamics among individuals performing the same task and present modelling results arguing that this similarity is not necessarily due to behaviour per se. This is important because they suggest that behavioural convergence can emerge from common underlying latent dynamics in the dynamics of anatomically different circuits. It is notable that the authors have arrived at their conclusions using only linear methods like CCA with samples from different models and regions. I am enthusiastic about the study and support its publication.

We thank the reviewer for their enthusiastic reaction about the importance of our work and their insightful comments.

However, two issues and several minor clarifications need to be addressed to better understand their findings and further strengthen the manuscript.

1. Latent dynamics and circuitry considerations

The authors could comment and analysis how latent dynamics relates to circuitry details. As the authors mention, low-level details of the circuitry do not necessarily account for behavioural similarities, however, for common latent dynamics to emerge there must be some structural similarities underpinning it – e.g., connectivity rank in the case of seminal work by Ostojic et al. The authors address this by training two RNNs to have dissimilar latent dynamics. However, it would be nice to conduct cross checks with controlled levels of latent structure dissimilarity (0, 30, 50 or 90%) to see how similar or dissimilar would the circuitry be then. Operating within their RNN model, one could modify the alpha parameter from equation 4 to see how it affects the underlying connectivity– i.e., different penalties on the CCA correlations.

Alternatively, authors could comment on how the similarity in the latent structure of motor cortex compares to the variability across individuals in, for example, their connectome (see Individual variability in functional connectivity architecture of the mouse brain, Communications Biology, 2020) or cell-type (see Phenotypic variation of transcriptomic cell types in mouse motor cortex, Nature, 2020). This could further improve the argument on how neural modes correlation is high in spite of variability in the underlying circuitry. Some quantification along those lines could be useful to reinforce the message about circuitry consequences.

While the phenotypic relation to circuitry is highly interesting, it is challenging to directly address with our current datasets. However, we agree that our modelling approach can be employed to further explore the link between connectivity structure and alignment. We find the reviewer’s suggestion to vary the alpha parameter and link the results with the RNN connectivity to be a very interesting idea. We have performed a new set of simulations in which we varied the value of parameter alpha thereby manipulating the extent of preservation of latent dynamics with respect to a standard (‘vanilla’) RNN (Figure R1A). Even if all subsets of networks generated very similar behaviour (Figure R1B-D), they exhibited distinct latent dynamics even after alignment (Figure R1E). The degree of dissimilarity between the latent dynamics in the networks was indeed controlled by varying parameter alpha in the cost function (Figure R1E): higher values of alpha were related to less preserved latent dynamics. When we reverse-engineered

the weights of the different networks, we found that more dissimilar latent dynamics corresponded to larger changes in the dimensionality (Figure R1F) and variance (Figure R1G) of the weights. This supports our hypothesis that preservation of latent dynamics is not just a trivial consequence of behavioural similarity; instead, it also likely reflects fundamental organisation and constraints in the underlying circuit implementation. We have now added this new analysis as Figure 5 in the revised manuscript.

Figure R1. CCA between different neural network architectures (LSTMs and RNNs) performing similar behaviour. **A.** We trained two sets of recurrent neural network models to perform the monkey centre-out reaching task. The first set of models were standard recurrent neural networks, while the second were constrained to produce distinct latent dynamics from the standard networks. The weight of this additional constraint was controlled by parameter α . **B-D.** Both standard and constrained networks produced similar behavioural output, as shown by the error (Panel C) and the correlation between 'reach trajectories' across networks (Panel D). **E.** Increasing the value of α decreased the preservation of latent dynamics, establishing that behavioural similarity is necessary but not sufficient to have preserved latent dynamics. **F-G.** Networks with distinct latent dynamics show differences in underlying connectivity. The dimensionality (Panel F) and the variance (Panel G) of the weight changes increased as the latent dynamics were constrained to be more distinct from those of standard networks by increasing α . Lines and shaded area, mean \pm s.d. across ten comparisons.

2. Analysis of the error trials in planning or covert behaviour

In both covert and overt behaviour, the planned and actual motion corresponds to a limited action space that has as its downstream output a concrete number of motor neurons. Hence, for a successful trial in the reaching task, future action plans must somehow converge even if the behaviour is a covert process. Is there any evidence of representations diverging if the intended movement does not coincide with the executed one? In other words,

whether error trials can be decoded and, in such case, if the motor representations between individuals are correlated. This could constitute more convincing evidence of how planning is shared, and it is due to common latent dynamics.

We think that this is a very interesting idea. Unfortunately, our existing data does not provide enough error trials to directly test since the monkeys were essentially performing at expert level with few mistakes, and the mouse task did not have an imposed preparatory period which would allow for careful study of the covert planning dynamics. However, we strongly predict that representations which are highly alignable during planning would diverge across individuals during execution if the intended movement did not coincide with the executed one. This is due to the tight link between the preservation of neural dynamics and behavioural output, as explored in the original manuscript in Figure 2K.

In case error trials are not readily available, some brief discussion would help regarding the embodiment of the problem – i.e., how activity convergence may be due to the recruitment of the same motor neurons – or in what regions representations may be more divergent – e.g., state or choice maps.

This is indeed a very interesting point, and we modified the Discussion in the revised manuscript to address how different regions could show different amount of preservation. In brief, we think that the preservation of latent dynamics across animals will decrease as one moves away from those regions more directly involved in producing behaviour (e.g., primary motor cortex or even dorsolateral striatum) or sensing the state of the body or the world (e.g., primary somatosensory cortex, and primary visual cortex). This prediction is based on the observation that “higher brain regions” such as prefrontal cortex (PFC) are dominated by large signals that relate to the internal state of the animal (Cowley et al, *Neuron*, 2020). Thus, even if two different animals are proficiently performing the same overt behaviour, the differences in their internal state (motivation, satiation, attention) would make specific components of the PFC latent dynamics different and thus not amenable to alignment.

3. More discussion on why striatum and motor cortex are dissimilar

The authors studied the latent dynamics both in the dorsolateral striatum of mice and M1 or PMd of monkeys and found that CCA improves predictive accuracy (Fig. 3) but much less effectively for striatum compared with M1 (Fig. 3 C,F).

We would like to clarify that since Fig 3F quantifies monkey motor cortex during covert behaviour, the most appropriate comparison for the striatal data of Figure 3C would be what is now Supplementary Figure S8B, which summarises decoding from mouse motor cortex during the same trials as former Figure 3C (now Figure 4C). However, we acknowledge that comparing these two different figures is not easy. To facilitate direct comparison between the two regions, which is a very interesting analysis, we have added a new panel to Supplementary Figure S8 (shown in Figure R2 below). As shown in this comparison of decoding accuracy for the same pairs of sessions from different individuals, the decoding performance after aligning motor cortical latent dynamics and striatal latent dynamics is virtually the same.

Figure R2. Comparison between across-animal movement decoder from M1 and striatum for the mouse grasping and pulling task. Markers, individual comparisons between each pair of sessions from two different mice. Note that, overall, movement kinematics can be predicted equally well from each of these two regions.

I couldn't locate information about whether the blue and the green data points for mice are significantly different, could the authors add the corresponding stars for all points?

We added these statistical results to Figures 2 and 4 of the revised manuscript.

Additionally, there seems to be significant decoding drop relative to the M1 data from mice after alignment, could the authors try to unpack the origin of this? Is there less a less common latent space? Decoding within seems to show a high R2 (0.6) but something fails for the CCA alignment in the striatum? How similar are trajectories (a visual would be helpful in the supplement)? Does the region code for something that is not strictly necessary for the same grasping motion and therefore is different across individuals? Does not align well across animals but aligns within the same animal on different trials? Is it a failure of the transfer to a different animal or poor alignment quality that could be better trial-to-trial within the same animal, a trial-vs-animal comparison would be helpful. Is there may be some additional normalisation needed, for speed or starting point to control for mouse-to-mouse behavioural differences? In any case it is interesting to comment more on why one would expect similar or dissimilar latent dynamics in the striatum compared to M1, its pathway position etc.

The reviewer raises a number of interesting points about the nature of the motor “representations” in M1 and striatum. While example latent trajectories illustrating the similarities in latent dynamics across these two regions were included in Supplementary Figure S6A,B of the original manuscript, we have now performed more direct comparisons between areas suggested by the reviewer. As shown in Figure R2 (above), our ability to decode across individuals from striatum and M1 are largely similar. Intriguingly, Figure R3 (below) shows that our ability to align striatal dynamics across mice is similar to that of M1. We included these two new figures comparing M1 and striatum as new panels in Supplementary Figure S8 of the revised manuscript.

Figure R3. Comparison between across-animal alignment from M1 and striatum for the mouse grasping and pulling task. Markers, individual comparisons between each pair of sessions from two different mice. Note that, overall, latent dynamics can be similarly well aligned across animals for each of these two regions.

Minor suggestions and corrections:

1) Predictive accuracy quantified through correlations (R^2), which is high if two variables evolve conjunctly in their dynamics but does not account for errors in distance between the predicted and true variables (L-norm). Supplementary Figure S5 shows traces are indeed similar in the primate case, one similar could be present for mice or could be shown by a RMSE plots. LSTM decoders may be fitting well the temporal dynamics, but the putative positions may have more errors in distance. How does the prediction fare in terms of distance metrics?

As per the reviewer's suggestion, we repeated the decoding analysis using a different Variance Accounted For metric which is sensitive to errors such as offsets in the predictions. The example in Figure R4 shows how the relative performance of within-animal and across-animal decoders is virtually unchanged compared to our original R^2 metric. We have included the quantification of decoder accuracy using the Variance Accounted For metric in Supplementary Figure S5 of the revised manuscript.

Figure R4. Decoding of hand position based on the latent dynamics using VAF as performance metric. Decoders trained on the aligned latent dynamics from one monkey allowed predicting hand position from a different monkey (blue) virtually as well as decoders trained and tested within the same session (grey), while decoding was poor without alignment as measured by both our original correlation-based metric (A), and a Variance Accounted For metric (B). Panel A reproduces Figure 2H). Circle and error bars, mean \pm s.d. across all monkey centre-out comparisons.

2) There are cases (Figure 2H or S4) where the across prediction is better than the within ones. In which cases the canonical space is benefitting the decoder?

While there are some spurious cases where the across predictions outperform the within predictions, these are not the general trends within the data. In all cases where the across predictions outperform the within predictions, the within predictions are abnormally poor (much lower than the rest of the data points). We hypothesise that the predictions are disrupted by idiosyncratic features in the population recordings such as nonstationarity (drift, etc) which provide a challenge for the multi-fold cross validation within animals. The alignment to other individuals actually can improve predictions by learning to ignore these features which are not directly related to the ongoing behaviour to be decoded, since such idiosyncratic features would not be readily aligned in the leading dimensions of both individuals.

3) Figure S3 shows that correlations are, to some extent, invariant to the manifold dimensionality. But do these results hold for low-rank cases (1-7 PCs)?

We have now performed this analysis. For these lower-rank cases, alignment is somewhat reduced which indicates that we are missing crucial information to fully reconstruct the dynamics that are preserved across individuals. Importantly, our ability to align the dynamics plateaus after 8–10 dimensions are considered in monkeys (Figure R5). This is notable because it suggests we are bound by the actual similarity in the latent dynamics, and are not susceptible to “overfitting”, in a sense, by overestimating

the dimensionality of the latent space. We have updated the figure in the manuscript (now Supplementary Figure S4B) to include the low-rank cases.

Figure R5. *Preservation of latent dynamics across all datasets for increasing neural manifold dimensionalities.* This plot shows the mean across the top four CCs (except when considering two and three-dimensional manifolds, in which case it is the mean across all dimensions) for the monkey movement (blue) and planning (purple) datasets (legend). This control further confirms that our results are robust across a wide range of parameters.

4) Correlations are shown by neural modes (Figure 2G, J and 3B-E), however, figures S1C and S5C present some diversity in the dimensionality of each individual by the number of components needed for a similar cumulative variance. Since $n=10$ is used for all post-CCA analysis, a concern is whether this divergence of dimensionality affects the analysis. Internal dynamics may be shared among individuals and be highly correlated or explanative of the variance but uninformative in terms of the behavioural readout. Plotting how decoding accuracy changes with a lower number of PCs could show this.

This is also an interesting analysis which is tightly related to the previous comment. Figure R6 below shows the outcome of this analysis for the same example session used throughout the paper and demonstrates that, considering as few as three dimensions yields a decoding accuracy that matches within-animal performance. We have included this analysis as a second panel in Supplementary Figure S5.

Figure R6. *Decoding of hand position based on the latent dynamics for manifolds with increasing dimensionality.* Data for all the monkey centre-out sessions. Line and shaded area, mean \pm s.d.

5) Errata: Figure S8 reads “similar to Figure 2I” where it should say Figure 2K instead.

Thank you for pointing out this error. We addressed it in the revised manuscript.

Referee #2 (Remarks to the Author):

This paper provides an analysis of neural population recordings from monkey and mouse brains which shows that the brains of different animals of the same species use similar dynamics to perform the same task or behavior. This is a remarkable finding given that the brains of individual animals have different wiring diagrams, cell type densities, and that only a small subset of neurons in each brain or brain region could be recorded. (The converse was not true: the dynamics of monkey brains while performing a reaching task did NOT resemble the dynamics of mice performing a grabbing-and-pulling task). The authors used PCA to identify the neural manifold (low-D subspace of neural activity) in each recording, then used canonical correlation analysis (CCA) to align the latent dynamics from different brains. Moreover, they show that a decoder trained on the aligned neural data from one brain can be used to decode behavior from the aligned neural data of another brain. The paper is well written and likely to be of interest to a broad readership.

We are very pleased to hear the reviewer's enthusiasm about our work.

My biggest concern is about how surprising it is that CCA turns up similar dynamics from different individual brains. The paper compares the observed similarity with a null distribution based on shuffled data. (The Methods section is a bit sparse and I admit I did not fully understand the shuffle control, which refers to using a correlations computed using a "randomly selected control window").

We apologise for the confusion. In the revised manuscript, we greatly expanded the details of the Methods section to satisfy the concerns of this reviewer and reviewer 3 below.

Regarding the control analysis specifically, in brief we randomly selected continuous windows of latent dynamics outside the epoch of interest (preparation or execution, depending on the aspect of behaviour we investigated in each analysis), concatenated them, and then tried to align these new data matrix to the actual neural data formed by all the concatenated trials. We chose this approach because these "control windows" have realistic temporal dynamics in the neural population activity—as opposed to pure random noise—such that if there were any low-dimensional structure in the neural data which could be found by chance within the actual population dynamics, the shuffle control could capture that.

However, I think it would be worthwhile to understand whether the level of correlation observed after CCA is something that would arise from any set of neural activities with the same temporal correlations and neuronal correlations. The most elegant (in my view) treatment of this issue came from a paper from Gamal Elsayed and John Cunningham:

Elsayed, G. F. & Cunningham, J. P. Structure in neural population recordings: an expected byproduct of simpler phenomena? *Nature Neuroscience*, 2017

In that paper, the authors set out to test whether the "rotational dynamics" reported in Churchland et al 2012 [ref 42] were truly surprising, or if they could be captured by any population that had the same temporal correlations, neuron-neuron (pairwise) correlations, and neuron-task correlations. They came up with a maximum-entropy model for generating new datasets that preserve those sets of pairwise statistics, and which could be used for testing whether the neural data from Churchland et al had rotational dynamics above and beyond that captured by the maximum-entropy null model.

Elsayad and Cunningham have released the code to fit their max-ent distributions to neural data, and so it seems like performing a similar test is something the authors of this paper could do without an excessive amount of additional work. If they find that the alignment of the neural data across monkeys is greater than that achieved by the max-ent distributions preserving the temporal, pairwise, and task correlations, then I will be substantially more convinced that the result is not a trivial consequence of lower-order statistical structure in the data.

(Actually, I am undecided about the necessity of including the neuron-task correlations in the null model; certainly it is a stronger result if the neural data also exceed the alignment of this null distribution, but at least I would think it should at least be more significantly aligned than the temporal and neuron-neuron correlations null model).

We agree with the reviewer that the Tensor Maximum Entropy (TME) method is an elegant approach to generate null distributions to check underlying hypotheses about neural population analyses—in fact, we have used it for several applications in our previous work (e.g., Gallego, Perich, et al, *Nature Neuro*, 2020; Gallego-Carracedo, et al, *eLife*, 2022). Implementing TME on this data revealed that the across-animal correlations in latent dynamics are well above the significance threshold derived from surrogate distributions that preserve the covariance over time, across neurons and across targets Figure R7. The same was true for all the conditions. Thus, our results are not a trivial consequence of lower-order statistical structure in the data. We included the TME control comparison in Supplementary Figure S2 in the revised manuscript.

Figure R7. Preserved motor cortical latent dynamics across monkeys largely exceed a new TME control that preserves all second order statistics of the data. **A.** Element-wise conservation of the covariance across time, targets and neurons between actual and surrogate data. **B.** Similarity in latent dynamics across all pairs of sessions from three different monkeys. Data include 21 sessions and 126 pairs of comparisons. Note that the correlation between aligned latent dynamics across monkeys (red) was quite close to the within-monkey correlations (grey), and largely exceeded both our original lower-bound control (orange) and the new TME control (mustard). **C.** Same as B but for mouse motor cortex during reach to grasp. **D.** Same as B but for monkey motor cortex during covert movement planning. Line and shaded area, mean \pm s.d. Monkey and Mouse illustrations by Carolina Massumoto

Another idea that occurred to me while reviewing the paper: what if the authors sought to align the data across monkeys with a shift in the labeling of the targets? That is, use CCA to align the datasets so that the neural data from reaches to target 1 in monkey A are aligned with neural data from reaches to target 4 with in monkey B. And then align target 2 data in A with target 5 data in B, target 3 with 6, 4 with 7, etc. Are the datasets still aligned if we are allowed to freely rotate the targets in one monkey compared to the other? This also would suggest that the alignment in the dynamics is unique to the specific behaviors, but also holds across shifts or permutations of those behaviors, which is a somewhat weaker result.

This is another interesting suggestion. We have explored this question using a new dataset, which provides much more behavioural complexity. Specifically, in order to increase the number of labels (that is, the number of targets) and help reduce confounding symmetries in the behavioural output that are present in the centre out task, we have now included a new “random walk” dataset from a previous paper (Glaser, Perich et al *Nature Comms* 2018). In brief, in this task monkeys had to perform a series of four continuous reaches to randomly generated positions within the workspace; only the next target in the sequence was presented at any movement. This led to considerable behavioural variability, yet, we could

match conditions across tasks (details in Figure R13 below) to generate as many as 29 conditions in the example dataset. Notably, aligning latent dynamics across reaches after permuting the labels led to correlations that were much lower than those obtained with matched labels, and that barely exceeded our new lower bound TME control (Figure R8). We included this analysis, along with a full demonstration and exploration of the random movement dataset, as new Figure 4 and Supplementary Figure S11 in the revised manuscript.

Figure R8. Permuting target labels across reaches prevents the alignment of motor cortical latent dynamics across monkeys performing a sequential reaching task. Red: without permuting the targets; Blue: after permuting; Orange: Null distribution; Dashed grey lines: within monkey upper bounds.

Or, what about the alignment between cortical and striatal data? Are the dynamics more similar within brain regions than across different brain regions? The results will again be more surprising if we find that the motor cortical dynamics of one animal are more similar to the motor cortical dynamics of a second animal than they are to striatal dynamics in either animal. (Although if not this would also tell us something useful).

This is an interesting question with a deep possibility for analysis. In fact, we are actively working on another manuscript to study cortico-striatal interactions during the generation. Figure R9, below, shows the result of this analysis. M1-striatal latent dynamics are more aligned within animal than each of the M1 and striatal dynamics are across animals. We believe this better alignment to be driven by the fact that, even if the consistency of the overt behaviour is the same within and across animals (as is the case here; compare Figure 2K in the original manuscript to Supplementary Figure 6B), their internal states (e.g., engagement, motivation and reward) are likely to be different. These internal states affect the ongoing neural activity across many regions simultaneously (Hennig et al, *Nature Neuro*, 2021; Levy et al, *Neuron*, 2020), which would improve our ability to align across regions within-animal on a given session.

Figure R9. Comparison of preservation of M1 and striatal dynamics across mice with the similarity in M1 and striatal dynamics within the same animal. Canonical correlation curves for 10-dimensional spaces after aligning across individuals with M1

(yellow) or striatum (orange) compared to aligning between M1 and striatum within an individual (dark grey). Line and shaded area, mean \pm s.d.

At this time, we are not intending to add this analysis to the revised manuscript as we feel such across-region comparisons are outside the scope of our original paper. Indeed, we think they are so interesting that they deserve a full exploration in the follow-up paper, and we include this preliminary analysis here for the reviewer's interest.

Other comments:

1. CCA returns a set of axes ordered by the strength of correlation, and these (descending) correlations are plotted, for example, in Fig 2F,G, and J. However, what seems to be hidden here is how much variance in the original manifold these CCA axes account for. That is, to what degree are the dominant correlated directions aligned with the major principal components of the data? If the 3 most correlated axes of the aligned data are loading mostly on the bottom 3 Principal Components, while the bottom CCA axes have high correlation with the dominant PCs, then this tells us that the alignment is not *that* strong --- i.e., the alignment only holds if we squint and look at the principal component axes with smallest variance. I suspect that is not the case, but it would be nice to quantify this alignment one way or the other. (I'm not sure exactly what metric to use, but at the very least we could look at what the PC projection of the top CCA axes is, or try to quantify how much variance is captured by the top CCA axes compared to the top PC axes). I apologize if the authors addressed this and I simply missed it!

We have now performed this analysis adopting a method that we developed for this purpose in (Gallego et al *Nature Comms* 2018). The result, shown in Figure R10, indicates that for the monkey reaching data the top four CCA axes explain ~15–45% of the total neural variance (that is, of the high-dimensional single neuron firing rates), while the top four PCA axes explain ~25–65%. Thus, we can be confident that the aligned latent spaces capture dominant features of the neural population activity. We included this result as part of Supplementary Figure S4.

Figure R10. Comparison of the total neural variance explained by the top four PCs and the top four CCs. Individual markers, individual comparisons for all monkey movement datasets.

2. One additional methodological concern I had was about the decoding of aligned data. To avoid double-dipping, the authors should compute the alignment from a training set, fit the decoder on this training set (or another training set, if desired), and then perform decoding on a held-out test set that was not used for computing the alignment (or fitting the decoder). From the methods section, it was unclear to me whether this was what was done or not.

We agree that this is a stronger demonstration of our method, though we chose our original implementation because of a trade-off with the number of available trials for training and testing the decoder. We performed this analysis to align M1 latent dynamics as monkeys performed the centre-out reaching task and the results remain largely unchanged (Figure R11). We included this result as part of Supplementary Figure S5.

Figure R11. *Cross-validation of both decoding and alignment.* Prediction accuracy of hand kinematics in the monkey dataset (R^2) using three sets of trials for training and testing (one for alignment, one for decoder training, and one for testing) to ensure full cross-validation. We obtained similar performance for the across and within animal decoding. Individual lines, each monkey comparison during movement execution. Circle and error bars, mean \pm s.d. across ten comparisons.

3. A more general comment is that I found the Methods to be quite sparse in general. There are a number of results that are described as "cross-validated", but it is unclear exactly what this means -- i.e., how the train-test splits were chosen, and exactly what was done in each split. Likewise, I found it hard to understand what some of the controls were, or how the null distribution was constructed in some cases. I would suggest a careful rewriting of this section with a lot more detail (i.e., including things that may seem obvious to the authors, but will not be obvious to all readers!)

Thank you for the suggestion. We agree that the Methods could be expanded, and we have greatly improved their clarity and level of detail in the revised manuscript.

4. Another paper worth citing is this one from Kenneth Latimer:

Low-dimensional encoding of decisions in parietal cortex reflects long-term training history

Kenneth W. Latimer, David J. Freedman

doi: <https://doi.org/10.1101/2021.10.07.463576>

This paper reports a finding of different dynamics in the same task if animals have a different training history. This does not directly contradict the results in this paper (but it is suggestive that the similarity found here *might* rely on the similarity of training histories). It seems worth citing, at least!

This paper is certainly relevant, and we added a citation in the revised manuscript.

5. Pg 7, line 115: "higher-order animals". I'm not sure it's still common to refer to species as higher and lower order; it might be worth finding a different adjective!

We adjusted our phrasing in the revised manuscript to say "behaving animals", as we expect many species along the phylogenetic tree to engage in these behaviours.

6. pg. 15: 297 The R^2 value, defined as the squared correlation coefficient between actual and predicted hand trajectories, was used to quantify decoder performance.

I don't believe it is standard to report R^2 as the squared correlation coefficient, which can produce artificially rosy results (since the correlation coefficient is insensitive to errors in the mean and scale of the prediction). I prefer --- and I believe it is standard practice as well --- to report R^2 as the coefficient of determination, defined as $1 - SSR / SST$, where $SSR = \text{Sum of squared residuals}$, i.e., $1/N \sum (y_i - \text{ypred}_i)^2$ and $SST = \text{sum of total squares}$, i.e. $1/N \sum (y_i - \text{mean}(y))^2$.

(For more see https://en.wikipedia.org/wiki/Coefficient_of_determination).

We appreciate the suggestion, which is similar to the request made above by reviewer 1. While there are many formulations of R^2 , it is true that our original formulation was not sensitive to errors in mean and scale as we were primarily interested in comparing the temporal dynamics. In the revised manuscript, we added a new figure (Supplementary Figure S5D) demonstrating the decoding performance with a new metric (Figure R12) to confirm our models also generalise when considering offsets and scale.

Figure R12 - Reproduces Figure R4 above. *Decoding of hand position based on the latent dynamics using VAF as performance metric.* Decoders trained on the aligned latent dynamics from one monkey allowed predicting hand position from a different monkey (blue) virtually as well as decoders trained and tested within the same session (grey), while decoding was poor without alignment as measured by both our original correlation-based metric (A), and a Variance Accounted For metric (B). Panel A reproduces Figure 2H). Circle and error bars, mean \pm s.d. across all monkey centre-out comparisons.

Referee #3 (Remarks to the Author):

This study by Safaie et al examines an interesting and important issue of systems neuroscience, namely whether some invariance —although it is not described as such here— across time within individuals, and across individuals, can be detected in the activity of circuits involved in the planning and execution of behavior. The approaches center on the extraction and comparison of “latent dynamics” from multi-neuronal recordings taken within and across individuals in two sets of data/experiments (each in one species: NHPs and mice). The conclusions are that such similar brain (cortical and subcortical) latent dynamics appear to be detectable in and across individuals for several sets of movements. Whereas the topic is really interesting, I was not convinced by the study (premise, data, and analyses presented in the manuscript) as it stands.

We are glad to hear the reviewer is interested in our work, and appreciate the clear guidance. We have carefully addressed each of these suggestions, and we hope that our clarifications in this rebuttal, together with our new analyses, and the revised manuscript adequately address the reviewer’s concerns.

Premise and hypotheses.

“Preserved”. The point is minor but confusing: the authors use this term throughout including the title when I think they might mean “similar” or “invariant”. Preserved (conserved would be equivalent) implies (to me at least) a common origin or common source (developmental, evolutionary etc). If it is what was meant, then I failed to see the justification.

Lines 5, 25, 34. “Idiosyncratic” neural circuitry.

Line 23: “large differences in brain circuitry”

The premise of the study seems to be the assumption that the brains of different individuals are entirely different from one another. But we know this not to be true, at some degree of organization and description: cell types are many but reliably identifiable across individuals in a species; with the exception maybe of central olfactory circuits (and even there, evidence for lack of organization is not always convincing), maps, mappings and connectivity conform to known and predictable patterns (eg, within and across layers, within and across areas etc) with a great degree of graph precision (eg, per cell type); they are, like neuronal identities, the results of precise developmental programs, followed by local instructions and refinement via plasticity. Circuits are therefore far from being idiosyncratic —except if one thinks of circuits as composed of identifiable individual neurons, which they are probably not in NHPs and rodents; the real issue here is therefore one of resolution and granularity of the description: at an appropriate scale, circuit designs are not idiosyncratic.

We appreciate the thoughtful feedback on our choice of words. The reviewer’s comments in this regard have helped us clarify this key point and provide a clearer overview of our premise in the Introduction and Discussion of the revised manuscript.

We chose the word “preserved” intentionally, as we, like the reviewer, feel that the word implies a common origin of the dynamics across the individuals. As stated in our manuscript, we posit that the preserved latent dynamics across individuals come precisely from the common low-level circuit constraints that the reviewer refers to. Indeed, in the reviewer’s comment regarding Line 23, they mention that the neural circuitry of different individuals arises from developmental programs which are “preserved” across those individuals. These are precisely the types of common origin constraints which led us to use the word “preserved”.

To reiterate and clarify our overarching logic: at some level of description, neural circuits in the brain are preserved—i.e., similarities arise from a common evolutionarily-specified developmental program—across individuals from the same species. These developmental “rules” are instantiated within each individual to give a unique set of individual neurons, with their own connections, etc. We refer to this

instantiation in each individual as “idiosyncratic”. Despite this idiosyncratic, individual-specific instantiation—which leads to circuits with different cell identities, as the reviewer says—, we identify similar latent dynamics when each individual performed similar behaviour. So we used “preserved” to refer to the similar latent dynamics because, in our interpretation, the alignability of latent dynamics across individuals stems from the preserved circuit properties governing the neural networks that gave rise to those latent dynamics.

While there remains much future work to be done to clarify the relationship between these circuit rules and the resulting neural population dynamics, we feel that making this initial link is justified. We have three analyses to provide support for the interpretation: **1)** within the idiosyncratic circuitry of each individual, latent dynamics are less similar when performing different behaviours than they are across individuals (see Figure R20 below and the response to the corresponding comment for more details on the similarities across these behaviours); **2)** the RNN control in the original manuscript demonstrated that networks with different cost functions performing the task equally well have dynamics that are much less similar than those of different networks with the same architecture; **3)** we have added an additional RNN control showing that we could create pairs of models that generated highly similar outputs with progressively more distinct latent dynamics by manipulating a parameter of the cost function. Reverse-engineering the weights of the different networks revealed that more dissimilar latent dynamics corresponded to larger changes in the dimensionality and variance of the weights (see Figure R1 above).

We hope that our logic is more clear to the reviewer about our specific choice of words. We expanded on these considerations in the Abstract, Introduction and Discussion of the revised manuscript to improve its clarity.

To some extent, the fact that one can find similar rules of organization in say, visual or motor cortex from one individual to another, using massive under-sampling of those circuits (typically a tiny fraction of the neurons present), already indicates widespread regularities and invariances that one could expect to find also in low-dimensional projections. So the real problem, it seems to me, is one of resolution (of recorded data, and of output reconstruction and prediction). I imagine for example that if the primate motor task in the present ms. consisted in distinguishing 100 or even 50, rather than 8 targets, accuracy would drop dramatically, neural data would need to be increased equally dramatically, and alignments of datasets might not be very good.

This is a very interesting line of questioning, and we agree that there should be a relationship between the dimensionality of the underlying neural manifold, the complexity of the behaviour, and the number of neurons that need to be recorded in order to sample it (as elegantly discussed, for example, in Gao et al, *bioRxiv* 2017).

We have added new data and analyses to directly explore this relationship and address the reviewer’s comment. First, we took motor cortical recordings taken during a sequential random walk reaching task (data from Glaser, Perich et al, *Nature Comms* 2018; Lawlor, Perich et al, *J Comp Neurosci* 2018). In this task, the monkeys made a continuous sequence of reaches to targets at random locations in the workspace (Figure R13A). This gave a highly complex and continuous distribution of reaches with different lengths, speeds, starting positions, and directions (Figure R13B). From these distributions, we matched reaches of similar direction across individuals, generating as many as 29 different conditions (examples in Figure R13C and Figure R14A). Even with this large number of conditions, we could align the latent dynamics across monkeys comparably to the 8-target centre out task (Figure R13D,E).

Next, by subsampling from the larger number of conditions, we could directly quantify the relationship between canonical correlations for neural latent dynamics across monkeys to the number of conditions in this task (Figure R13F; Figure R14B shows all the canonical correlations for the example comparison).

A crucial result is that for all sessions these curves begin to saturate after ~ 9 conditions. In other words, for the same number of neurons and the same number of trials per condition, increasing complexity (here, the number of conditions) does not impact our ability to align the neural latent dynamics.

Figure R13. Preserved motor cortical latent dynamics across monkeys engaged in up to 29 reaching conditions in a sequential reaching task. **A.** Monkeys were trained to perform sequences of four reaches to randomly-placed target locations using the planar manipulandum. **B.** Example hand positions in the workspace for all reaching movements made during a session from two different monkeys. **C.** Example matched reaches across two monkeys in sessions with 28 reach conditions. Colours, different conditions. **D.** The aligned latent dynamics across monkeys are very close to the upper bound defined by comparing reaches from the same 28 conditions within the same monkey. **E.** Preservation of latent dynamics across pairs of monkeys performing matched targets during a sequential reaching task. Line and shaded area, mean \pm s.d. **F.** The preservation of latent dynamics holds for a wide range of conditions. The plot shows, for each pair of sessions, the mean of the top four canonical correlations between the latent dynamics as a function of the number of conditions subsampled from the total available in each session. Data from 10 pairs of random-target sequential reaching sessions from three different monkeys. Trace in black indicates the example comparison shows in Panels B-D. **G.** Preserved latent dynamics can be uncovered from even sparsely sampled neural populations. The plot shows, for each pair of sessions, the mean of the top four canonical correlations between the latent dynamics as a function of the number of neurons subsampled from the total available in each session. Trace in black indicates the example comparison shows in Panels B-D. Monkey illustrations by Carolina Massumoto.

Finally, we included a new analysis that shows that our ability to align the latent dynamics is not dependent on the number of recorded neurons, provided that a sufficiently large population is considered. For the comparison with 28 conditions, the quality of the alignment obtained with latent dynamics computed from as few as ~ 30 neurons, was virtually identical to that obtained from 137 neurons (Figure R13G, in black; Figure R14C shows all the canonical correlations for the example comparison). We included these analyses (as well as the permutation comparison shown in Figures R8 and R14D) in two new figures in the revised manuscript: Figure 3 and Supplementary Figure S11.

Figure 14. Additional data related to the preservation of motor cortical latent dynamics across monkeys engaged in up to 29 reaching conditions in a sequential reaching task. **A.** Example normalised neural firing rates aligned to movement onset for two different monkeys (top) and corresponding hand trajectories (bottom). Each column, one of 28 conditions. Data from the same example dataset show in Figure R13B-D. **B.** Canonical correlations as function of the number of conditions considered (legend). Note that the traces line up very well after ~9 conditions are considered. Data from the same example dataset as in Panel A. **C.** Canonical correlations as function of the number of dropped neurons (legend). Note that the traces line up very well after ~57 neurons are considered. Data from the same example dataset as in Panel A. **D.** Permuting the condition labels (blue) considerably decreases the alignment of latent dynamics across animals, bringing the canonical correlations closer to the lower bound control. Data from ten comparisons across three different monkeys.

When in line 34-35, the authors say “...low level details of neural circuits ... should not be necessary to account for the emergence of species-typical behaviors”, I wonder what is meant. Everything is a question of degree and of precision. My ability to play the violin and the kinematics of our movements likely differ from Itzhak Perlmann’s by many low-level details of our respective neural circuits. At some level of description, we must diverge, but there may be low-d approximations where those differences cannot be detected. It is therefore a bit circular.

The issue of “nature vs nurture” is fascinating to us, and it’s interesting to think about the differences between development and learning. Indeed, we also believe that Itzhak’s extensive practice will have led to changes in motor cortical circuitry that we and the reviewer—not being able to judge the reviewer’s musical skills, but presuming they are not one of the world leading violinists—have likely not undergone. We think it remains an open question just how impactful these learning-induced changes in circuitry will be on the resulting neural latent dynamics.

If we imagine that a virtuoso plays a melody requiring movement patterns that we can’t produce (probably because of the changes in circuitry that result from extensive practice), we would not be able to directly compare the latent dynamics—our brain could not have produced those dynamics because we could not produce the behaviour. However, if we now imagine that the virtuoso then plays a simple melody that we can reproduce, and thus our behaviour is exactly the same, we can consider two hypotheses when comparing our neural latent dynamics to Itzhak’s.

Hypothesis A: Extensive rewiring for finger dexterity has sufficiently altered Izahk's cortices such that even when doing simple things like playing a single note (or, performing a centre out reaching task in the lab), the motor cortex would generate dynamics that are different from ours.

Hypothesis B: The refinements induced during practice entrains Izahk's cortices to be able to more accurately and efficiently produce the latent dynamics necessary for these movements, but the fundamental structure in these latent dynamics is unchanged compared to what we produce.

While we do not currently have the data to definitively address this question, based on our existing data and results we hypothesise that the second option is more probable (in part because the different monkeys had varying amounts of training, with some more expert than others) though this remains speculation for the time being. Regardless, it's an important and interesting point that we address in the discussion of the revised paper.

In addition, because it is already known that similarities can be found across recordings made during motor execution and during the preparation for intended movement in motor/premotor areas, the fact that invariances may be found for movement execution would seem to lead logically to the prediction that preparation for action should similarly see invariances across trials and across individuals. The opposite finding would have been interesting, because unexpected (I think).

We included this analysis for two primary reasons: 1) to show that our finding of preserved latent dynamics across individuals also applies to covert behaviours, where the neural latent dynamics could be primarily generated internally by other brain regions; and 2) to mitigate potential concerns about the preservation of latent dynamics during execution being largely driven by similar afferent feedback trivially resulting from the similar behaviour.

We agree with the reviewer that this result is consistent with a view in which motor cortical dynamics during preparation serve as initial conditions for the dynamics that evolve during execution. However, the ability to align during preparation is not trivially expected, as preparation (while related to future execution) is not subject to the same temporal constraints as the movements generated by overt behaviour.

Methods.

I found it hard to put together many of the methodological details that matter in the study.

We apologise for the confusion. In the revised manuscript, we greatly expanded the methodological details in all sections, with particular focus on the details described by this reviewer.

To start and most trivially, how many neurons were recorded in each species and experiment? Where they sorted single units? Multi-units?

While we neglected to include the neuron counts in the original manuscript, the original manuscript stated that the datasets were a mix of sorted single units and multi-units (copied below). We saw little difference between the two data types, and it has been demonstrated that both data types are adequate to estimate latent dynamics in neural population activity (see Trautmann et al, *Neuron*, 2019). We updated the revised manuscript to clearly state which datasets were multi-units and which were sorted single units, as well as the number of neurons in each individual (see Methods section).

Page 11: “We also manually spike sorted the recordings from monkeys C and M to identify putative single neurons. Monkey J had fewer well-isolated single units than the other monkeys, so rather than spike sorting we directly applied the multi-unit threshold crossings acquired on each electrode.”

Page 12: “Recorded data were pre-processed using an open-source software KiloSort 2.0 ... and manually curated using Phy ... to identify putative single units in each of the primary motor cortex and dorsolateral striatum.”

What datasets were from already published experiments, what were new ones?

All of the monkey data has been previously published, and many datasets are already publicly available as stated in the Data Availability statement following the Methods. The mouse data has never been published before, although other datasets from a previous version of this task were included in (Park et al., *Science Advances* 2022). For maximum clarity, we also state in the methods section describing the monkey dataset that the data were previously published.

Data were reduced to 10D by PCA, and this choice of 10 modes is not precisely justified, or the consequences of this choice explored (except around 10 ± 3 in one analysis). These 10PCs, one finds in the methods, explain about 60% of the data. Is that a reasonable representativeness to address the questions posed? What does the spectrum of eigenvalues look like?

We would like to note that the cumulative version of the eigenvalue spectra for all monkey and mice sessions were included in Figure S1 and S5 of the original manuscript. In this revision, we have further expanded the dimensionality analysis to cover lower-rank dimensions (Supplementary Figure S4B, partially reproduced in Figure R5 above), and we have added a new analysis to quantify the variance captured by the aligned canonical correlation dimensions (Supplementary Figure S4A, reproduced in Figure R10 above). Overall, all the core results—preservation of latent dynamics, behavioural relevance based on across-animal decoding accuracy, etc.—are robust across a broad range of dimensionalities, which indicates that we are capturing a fair representation of a robust phenomenon. .

Said differently, one would like to know at what degree of dimensionality reduction the equivalences or invariances (eg, between responses recorded from different individuals) stop being identifiable. Some of the plots (eg in Fig 2F and G) suggest that the dimensionality needs to be considerably reduced.

We agree that this is an interesting question. It is worth noting that the monkey centre out task is effectively ~ 2 – 3 dimensional. At or below this dimensionality of 2, we expect a mostly trivial amount of alignment within the population activity in the sense that the alignment could be explained by low-level behavioural features alone. The compelling feature of our data is that the neural alignment is much higher for many more dimensions above 2 (in contrast, for example, to the RNNs of Figure S10 of the original manuscript which align effectively for only 2 to 3 dimensions). In the revised manuscript, we explore this more fully by extending Supplementary Figure S3 (now Figure S4B)—as also requested by reviewer 1—to include a wider range of manifold dimensionalities (Figure R15). Our results show that for very low-rank cases (e.g., 2 or 3 dimensions) alignment is somewhat reduced which indicates that we are missing crucial information to fully reconstruct the dynamics that are preserved across individuals. Importantly, our ability to align the dynamics plateaus after 8–10 dimensions are considered in monkeys (Figure R15). This is notable because it suggests we are bound by the actual similarity in the latent dynamics, and are not susceptible to “overfitting”, in a sense, by overestimating the dimensionality of the latent space.

Figure R15 — Reproduces Figure R5 above. *Preservation of latent dynamics across all datasets for increasing neural manifold dimensionalities.* This plot shows the mean across the top four CCs (except when considering two and three-dimensional manifolds, in which case it is the mean across all dimensions) for the monkey movement and planning datasets (legend). This control further confirms that our results are robust across a wide range of parameters.

We complemented this alignment analysis with a decoding analysis (Figure R16; included as Supplementary Figure S5B in the revised manuscript). Intriguingly, the ability to decode saturated at a much lower dimensionality (as low as 2-3 dimensions) than the ability to align neural dynamics, even for comparisons across individuals. Thus, the alignment of neural latent dynamics across individuals is not trivially driven by the lowest-level behavioural similarities (e.g., limb kinematics).

Figure R16. *Decoding of hand position based on the latent dynamics for manifolds with increasing dimensionality.* Data for all the monkey centre-out datasets. Line and shaded area, mean \pm s.d.

Interpreting the purpose and utility of these higher-dimensional representations is an interesting but ongoing line of research for many labs (see, for example, the work from Mark Churchland and colleagues, such as Russo et al, *Neuron*, 2018). Thus, while it’s important to explore possible alternative dimensionalities (see comment above), we do not necessarily agree that the dimensionality should be reduced in our current dataset and analysis. On the contrary, the CCA algorithm is identifying alignable features in the dynamics of these higher dimensions.

Inter-individual reduced representations were aligned by CCA. But if the representation of angle (in the NHP pointing task) in neural space is topologically ordered within individual animals, alignment by CCA across animal representations should not come as a surprise, or should it?

This is a very interesting point. We have previously addressed in an earlier paper demonstrating the stability over time—but within individual—of the latent dynamics underlying the same behaviour (Gallego, Perich et al *Nature Neurosci*, 2020). There, we showed that aligning the “topological representations” of the neural data does not suffice to uncover stable latent dynamics or achieve accurate across-day decoding.

We have now reproduced this analysis on the monkey centre-out dataset to similarly establish that aligning the topological organisation across reach directions is not sufficient to identify preserved latent

dynamics across individuals performing the same task. We first averaged each trial over time to get a single data point that reflects the topological organisation of the neural activity (Figure R17A). We then applied CCA to align these topological structures across individuals (Figure R17B). Interestingly, when we used the resultant linear transformation (CCA) matrices to align the time-varying latent dynamics, we found that these transformed latent dynamics were much less similar than those obtained when using CCA matrices that also accounted for the dynamics (Figure R17C,D)—a much more dramatic effect than when studied the stability of latent dynamics over time within the same individual performing the same task on different days. In fact, the mean across the four top canonical correlations decreased dramatically. Therefore, one has to preserve not only the topological organisation, but also the temporal structure of the activity to uncover preserved latent dynamics. We have included this analysis as Supplementary Figure S10.

Figure R17. *Aligning based on the topological organisation of the neural data is not sufficient to uncover preserved latent dynamics across individuals.* **A** To align the latent dynamics across sessions based on the topological organisation of the population activity, we performed CCA on the time-averaged activity. **B.** This method provided excellent alignment of the topology on a single trial basis. Left: each point represents a reach to one of the eight targets (colour code in inset) during one session from Monkey C_L (closed circles) and Monkey M (open squares). Right: after alignment, similar target-specific structure is present across both animals. **C.** Pairwise comparisons of the CCs after projecting the latent dynamics onto the manifold axes found by aligning the topology (vertical axis) and onto the manifold axes found by aligning the latent dynamics (horizontal axis). Data shown for the top six neural modes (see legend for colour code). Each dot represents one comparison. All dots lie well below the diagonal (dashed grey), indicating that aligning the latent dynamics based on the topology does not reach the CC values obtained by aligning the latent dynamics. **D.** Correlation values were dramatically lower when the alignment was based on preserving the topological structure of the neural population activity rather than its latent dynamics, illustrating the importance of the precise temporal dynamics for uncovering preserved latent dynamics. For Panels C and D include data from 21 sessions and 126 pairs of comparisons during the monkey centre-out reaching task.

In the example with mouse data, the behavior is described as consisting of 4 tasks (two positions of a lever times two resistance levels), but the resistance level data for each position are merged in the analysis, making it in fact a 2-option task. It seems evident that aligning them across animals should not be difficult (the opposite would have been surprising). Hence the mouse data seem to provide very little of interest here.

We originally did this analysis in this manner because there was considerable within and across mouse variability in the behaviour, as shown in Supplementary Figure S6A,B. However, as per the reviewer's

suggestion, we have now considered all four conditions by focusing on the peri-pull window (-50 ms before pull until +300 after pull). These results, shown for M1 in Figure R18 show that we can still align the latent dynamics quite well, and significantly above our new, more principled lower bound control identified using the Tensor Maximum Entropy method (see comments by reviewer 2 above). We have included this new result in the revised manuscript as Supplementary Figure S8A.

Figure R18. Preserved motor cortical latent dynamics across mice performing the grasping and pulling task when considering four conditions. We repeated the alignment analysis using additional data taken from the mouse reach to grasp task. Here, we consider both the reach and pull periods as separate conditions, giving us four in total (2 directions x 2 loads) rather than two as in the original manuscript (2 directions). Our ability to align in the 4 condition case (red) is slightly increased compared to the 2 conditions case (blue). Upper and lower bounds are provided by within-individual alignment (gray) and the TME control (orange), respectively. Mouse illustration by Carolina Massumoto.

Controls.

I found it very hard to find the information concerning the controls (eg fig 2F, G etc). On line 58, the authors state a control with “randomly selected behavioral epochs”. In the methods’ Data Analysis section (line 228-231), however, the control windows are called “behaviorally irrelevant” windows, and defined as 450ms segments “taken randomly along inter-trial and trial periods combined”. What does that mean exactly? Why are they called behavioral epochs in the results and are not so in the methods? How are these windows constructed/chosen? Those details would seem to matter a great deal.

We apologise for the lack of clarity on some of the methodological details. In the revised manuscript, we greatly expanded the details of the Methods section to satisfy the concerns of this reviewer and reviewer 2.

Regarding the implementation of the control analysis in the original manuscript, in brief, we randomly selected continuous windows of latent dynamics outside the epoch of interest (preparation or execution, depending on the aspect of behaviour we investigated in each analysis), concatenated them, and then tried to align these new data matrix to the actual neural data formed by all the concatenated trials. We chose this approach because these “control windows” have realistic temporal dynamics in the neural population activity—as opposed to pure random noise—such that if there were any low-dimensional structure in the neural data which could be found by chance within the actual population dynamics, the shuffle control could capture that.

In the revised manuscript, we have also implemented a more principled control distribution by adapting the Tensor Maximum Entropy method (Elsayed & Cunningham, *Nature Neuro* 2017) as suggested by reviewer 2. TME is an elegant method to generate null distributions that preserve all second statistics of the original single neuron data to check underlying hypotheses about neural population analyses. Implementing TME on our data revealed that the actual across-animal correlations in latent dynamics are well above the significance threshold derived from surrogate distributions that preserve the covariance over time, across neurons and across targets (Figure R19A), which provides an additional

and hopefully more clear demonstration of the significance of our results (Figure R19B-D). We have included the new TME control as Supplementary Figure S2.

Figure R19 — Reproduces Figure R7 above. Preserved motor cortical latent dynamics across monkeys largely exceed a new TME control that preserves all second order statistics of the data. **A.** Element-wise conservation of the covariance across time, targets and neurons between actual and surrogate data. **B.** Similarity in latent dynamics across all pairs of sessions from three different monkeys. Data include 21 sessions and 126 pairs of comparisons. Note that the correlation between aligned latent dynamics across monkeys (red) was quite close to the within-monkey correlations (grey), and largely exceeded both our original lower-bound control (orange) and the new TME control (mustard). **C.** Same as B but for mouse motor cortex during reach to grasp. **D.** Same as B but for monkey motor cortex during covert movement planning. Line and shaded area, mean \pm s.d. Monkey and mouse illustrations by Carolina Massumoto.

Test in lines 85-89: In this comparison, the authors report that the latent dynamics across animals for the same task are more similar than within the same animal across tasks. Again, the devil is in the detail of the movements and behaviors. If the movements are completely different, engaging different muscles, joints and forces, would this not be expected?

Indeed, the reviewer’s logic is correct and we expect alignment to be worse if the movements are completely different. We included this analysis from (Gallego et al, *Nature Comms* 2018) because this is an important clarification of the importance of behavioural similarity for our across-individual results since in that paper we compared two sets of related tasks. The similarity of the behaviours that we consider in the comparison is indeed critical. The original paper provides all the details requested by the reviewer, which we summarise next. If we focus on the “wrist manipulation” tasks, in all cases the device and the target organisation was the same, and so was the configuration of the arm, which was restrained both in the upper and lower parts. Visual feedback was identical.

We now focus on one particular case, the comparison between a one-dimensional movement task and a one-dimensional spring-loaded movement task. These two tasks required producing the same one-dimensional movements, with the same amplitude, using the same manipulator, and while the arm was fixed in the same posture; the only difference was the velocity profiles when acquiring different targets. Consequently, the same muscles were engaged in the task (Suppl Figure 1 in the original paper). Moreover, we only considered units that we could track during the entire session, and in some cases their activity was quite similar across these two tasks. Likewise, the covariance activity patterns across the entire recorded population were also quite similar across tasks (Figure 3 in the original paper). Yet, despite all these similarities, the latent dynamics across these two quite similar tasks were lower than those reported here across monkeys but within-task (compare the mustard and red traces in Figure R20). This was the case for all the pairs of tasks we considered, which also included other wrist task comparisons (light grey traces in Figure R20), and two reach-to-grasp tasks (dark grey traces in Figure R20G; these tasks included reaching, grasping and carrying a ball, and reaching grasping and squeezing a manipulandum).

Moreover, our ability to decode across tasks (Figure 6 in the original paper) was much lower compared to our ability to decode across monkeys but within-task. Thus, we can conclude that behavioural similarity crucially determines our ability to align the latent dynamics. We have included this updated figure as Supplementary Figure S12 in the revised manuscript.

Figure R20. The latent dynamics of two different monkeys performing the same task are more similar than those of the same monkey performing two related wrist or reaching and grasping tasks. Despite the similarities in task characteristics (B), motor output, engaged muscles, and even single unit tuning, the latent dynamics generated by the same monkey performing different wrist tasks (mustard and light grey) or different reaching task (dark grey) within the same day were less similar than those of two different monkeys engaged in the same task (red).

And would it not be more appropriate to carry out CCA on (reduced) within-animal datasets recorded for the two behaviors, and then test across both animals and behaviors?

As we showed in the 2018 study referenced in the original manuscript (Gallego et al, *Nature Comms* 2018), different behaviours have both shared and unique aspects of the dynamics (Figure 5 therein). We can imagine, then, that there exists a continuum of behavioural similarity with a matching continuum of preservation of neural latent dynamics. We would thus hypothesise that any shared components between these behaviours would be alignable both across and within individuals, though fully testing this hypothesis would require datasets where we can construct a more densely sampled continuum of behavioural similarity.

In short, while I found the topic of the study extremely interesting, I was not convinced by the data, analyses and interpretations as they stand now.

We thank you for your interest in our study, and appreciate how the clear guidance helped us improve the manuscript. We hope that these comments help clarify our claims in the manuscript and provide sufficiently convincing evidence to support them.

References

- Cowley B.R., Snyder A.C., Acar K, *et al.* Slow Drift of Neural Activity as a Signature of Impulsivity in Macaque Visual and Prefrontal Cortex. *Neuron*. **3**, 108. (2020).
- Dekleva, B.M., Kording, K.P. & Miller, L.E. Single reach plans in dorsal premotor cortex during a two-target task. *Nat Commun* **9**, 3556 (2018).
- Elsayed, G., Cunningham, J. Structure in neural population recordings: an expected byproduct of simpler phenomena?. *Nat Neurosci* **20**, 1310–1318 (2017).
- Feulner, B., Perich, M.G., Chowdhury, R.H. *et al.* Small, correlated changes in synaptic connectivity may facilitate rapid motor learning. *Nat Commun* **13**, 5163 (2022)
- Gallego, J.A., Perich, M.G., Naufel, S.N. *et al.* Cortical population activity within a preserved neural manifold underlies multiple motor behaviors. *Nat Commun* **9**, 4233 (2018).
- Gallego, J.A., Perich, M.G., Chowdhury, R.H. *et al.* Long-term stability of cortical population dynamics underlying consistent behavior. *Nat Neurosci* **23**, 260–270 (2020).
- Gallego-Carracedo C., Perich M.G., Chowdhury R.H., *et al.* Local field potentials reflect cortical population dynamics in a region-specific and frequency-dependent manner. *eLife* **11**:e73155. (2022)
- Glaser J.I., Benjamin A.S., Chowdhury R.H., *et al.* Machine Learning for Neural Decoding. *eNeuro*. **7**, 4 (2020).
- Hennig, J.A., Oby, E.R., Golub, M.D. *et al.* Learning is shaped by abrupt changes in neural engagement. *Nat Neurosci* **24**, 727–736 (2021)
- Lawlor, P.N., Perich, M.G., Miller, L.E. *et al.* Linear-nonlinear-time-warp-poisson models of neural activity. *J Comput Neurosci* **45**, 173–191 (2018).
- Levy S., Lavzin M., Benisty H., Ghanayim A., *et al.* Cell-Type-Specific Outcome Representation in the Primary Motor Cortex. *Neuron*. **5**, 108. 954-971 (2020).
- Sussillo, D., Churchland, M., Kaufman, M. *et al.* A neural network that finds a naturalistic solution for the production of muscle activity. *Nat Neurosci* **18**, 1025–1033 (2015).

Reviewer Reports on the First Revision:

Referees' comments:

Referee #1 (Remarks to the Author):

The authors provided a very comprehensive response to my comments and have very much improved the paper and answered all the comments well. I have no further comments nor remaining suggestions for improvement beyond reducing the abstract length, if possible. It looks like the results are powerful enough to not require such a detailed introduction or motivation, in my opinion. I would like to congratulate the authors on such excellent and self-contained research and recommend publication.

Referee #2 (Remarks to the Author):

The authors have done a very thorough job addressing my comments and suggestions, and I thank them for their detailed replies and clarifications. I believe the paper is now substantially stronger and will make an excellent contribution to the literature.

Referee #3 (Remarks to the Author):

I thank the authors for the added data and clarifications. This has very much improved the ms and I find the results interesting.

The paragraph starting at L 175 is a nice attempt to incorporate my remarks but the reframing does seem a bit contrived (to me).

Despite this, my initial global problems with the results and interpretations remain:

1. what is the granularity needed to describe and differentiate behavioural outcomes within and across individuals, and at what point can the latent dynamics or low-D manifolds be considered to be same or different? Indeed, despite the very quantitative approach, the descriptions and conclusions—eg “dominant”, “robust over a broad range of dimensionalities”, “align quite well”, “quite similar across tasks”—seem very qualitative. The reader is offered 2d plots (corr vs neural modes) with pretty steep degradation, enabling the general conclusion that “across monkeys” is better than within and worse than TME controls. So the question becomes: is this surprising and how surprising is it? I found these questions difficult to answer.

2. the Shenoy 2019 paper analyses data in which 10-15 dimensions accounted for 90% of the variance; in the present paper, it is 60%, although I could not figure out if the data were pooled and analysed in the same manner. The 2019 paper showed, as the authors here cite, that sorted spikes (single units or SUs) and thresholded MUs yielded equivalent low-D geometries in motor cortex, while noting that this may not hold for other areas (eg sensory). I suppose that limb control in

primates (and probably, in general) is highly constrained by the biomechanics of the limb (few degrees of freedom compared with what probably describes the mobility of an elephant's trunk, a dog's tongue or an octopus's arm). In these NHP and mouse experiments, the movements are also constrained by manipulanda. Hence, the low dimensionality of possible behavioral outcomes, the low sensitivity of the latent geometry to the recording resolution (results with MUs = results with SUs), the apparent necessity that similar movements should be accompanied by reasonably similar neural dynamics (these dynamics after all, must explain temporal sequences of arm muscle activations), and the basic similarity of brain architecture across individuals of a given species (resulting from genetic and developmental programs + equivalent refinements due to learning) together seem to make the results somewhat predictable (at least, not entirely surprising at a suitably low level of resolution). It is nice to show it of course, but again, I do not know how to evaluate my surprise (or lack thereof).

3. How do the observations scale with the complexity of the task? The added random-movements experiment starts to address this, but what would it look like in a cortex (or other brain region) where encoding is more sensitive, where $MU \neq SU$, where behavioral or sensory space is not 2-3 dimensional? RNNs are not a terribly useful comparison for cortex or brains in general. Lacking this, it is difficult to know what to extrapolate from the results to other regions.

4. I now note that the primate data have appeared before in 6 publications. I know that this may be standard practice in this field, but should it be?

In conclusion, the paper has been very much improved, should and could appear as it is somewhere, but whether it is sufficiently compelling to appear in Nature is unclear to me.

Author Rebuttals to First Revision:

Response to referees' comments:

Referee #1 (Remarks to the Author):

The authors provided a very comprehensive response to my comments and have very much improved the paper and answered all the comments well. I have no further comments nor remaining suggestions for improvement beyond reducing the abstract length, if possible. It looks like the results are powerful enough to not require such a detailed introduction or motivation, in my opinion. I would like to congratulate the authors on such excellent and self-contained research and recommend publication.

We would like to thank the reviewer for their suggestions that have helped to greatly improve our paper, as well as for the kind words and enthusiasm about our work.

Referee #2 (Remarks to the Author):

The authors have done a very thorough job addressing my comments and suggestions, and I thank them for their detailed replies and clarifications. I believe the paper is now substantially stronger and will make an excellent contribution to the literature.

We would like to thank their reviewer for their appreciation of our work and their suggestions during the review process. We firmly believe that the paper is stronger as a result.

Referee #3 (Remarks to the Author):

I thank the authors for the added data and clarifications. This has very much improved the ms and I find the results interesting.

We are delighted to hear that the reviewer finds the manuscript to be improved and interesting.

The paragraph starting at L 175 is a nice attempt to incorporate my remarks but the reframing does seem a bit contrived (to me).

We do not view that paragraph as a reframing, *per se*, but more a restatement of the logic and conclusions we explore throughout the paper. We wanted to lay out the logic of our conclusions as clearly and simply as possible. We think it is useful for other readers to see the logic in our arguments as directly as possible.

Despite this, my initial global problems with the results and interpretations remain:

1. what is the granularity needed to describe and differentiate behavioural outcomes within and across individuals, and at what point can the latent dynamics or low-D manifolds be considered to be same or different? Indeed, despite the very quantitative approach, the descriptions and conclusions—eg “dominant”, “robust over a broad range of dimensionalities”, “align quite well”, “quite similar across tasks”—seem very qualitative. The reader is offered 2d plots (corr vs neural modes) with pretty steep degradation, enabling the general conclusion that “across monkeys” is better than within and worse than TME controls. So the question becomes: is this surprising and how surprising is it? I found these questions difficult to answer.

We agree that there is a very interesting line of inquiry to be found in the granularity necessary to describe similarities or differences across individuals. In brief, as the reviewer points out in Comment #2 below, the limb movements studied here are bound by biomechanical constraints (degrees of freedom and articulation, mechanical properties, etc) which help reduce the dimensionality of the space required to produce the movements. We think of this as somewhat of a “chicken or the egg” problem: these same biomechanical constraints also helped shape the organisation of the motor control circuits of the brain. In our particular case, the reduced dimensionality is advantageous because it allows us to compare low-dimensional dynamics across individuals. However, we can imagine different tasks such as odour discrimination in which the dimensionality of the space is less constrained by such intrinsic factors. Here, it may be necessary to look with a finer granularity in order to establish the preservation of neural latent dynamics. We think this is a fascinating question for future work, but we can speculate—based in part on our results in the more complex random movement task—that one would observe similarly preserved latent dynamics across individuals as we have in this paper provided the appropriate granularity is achieved and the sensation or behaviour is sufficiently similar. The key is finding an appropriate granularity to ensure you can capture the behaviourally relevant dynamics of interest, which will vary based on specific experiments.

Regarding the reviewer’s second point, we agree those statements are qualitative, but the quantitative alignment results are compared to several controls giving both upper and lower bounds to establish their relevance. We also want to point out that the analysis of decoding accuracy across animals directly quantifies the behavioural relevance of the preserved latent dynamics.

We have modified the second paragraph of the Discussion to address the granularity necessary in our task versus other potential behaviours or regions.

2. the Shenoy 2019 paper analyses data in which 10-15 dimensions accounted for 90% of the variance; in the present paper, it is 60%, although I could not figure out if the data were pooled and analysed in the same manner. The 2019 paper showed, as the authors here cite, that sorted spikes (single units or SUs) and thresholded MUs yielded equivalent low-D geometries in motor cortex, while noting that this may not hold for other areas (eg sensory). I suppose that limb control in primates (and probably, in general) is highly constrained by the biomechanics of the limb (few degrees of freedom compared with what probably describes the mobility of an elephant's trunk, a dog's tongue or an octopus's arm). In these NHP and mouse experiments, the movements are also constrained by manipulanda. Hence, the low dimensionality of possible behavioral outcomes, the low sensitivity of the latent geometry to the recording resolution (results with MUs = results with SUs), the apparent necessity that similar movements should be accompanied by reasonably similar neural dynamics (these dynamics after all, must explain temporal sequences of arm muscle activations), and the basic similarity of brain architecture across individuals of a given species (resulting from genetic and developmental programs + equivalent refinements due to learning) together seem to make the results somewhat predictable (at least, not entirely surprising at a suitably low level of resolution). It is nice to show it of course, but again, I do not know how to evaluate my surprise (or lack thereof).

*In the 2019 Shenoy paper, they analysed trial average activity, saying: “where roughly 10–15 dimensions capture 90% of the variability of trial-averaged neural population activity when monkeys perform a simple 2D reaching task.” In our paper, we analyse single trial data by concatenating all individual movements to test decoding and alignment. This greatly increases the total amount of variance to explain, which accounts for our need to increase the number of dimensions to reach the same proportion of variance explained. As the reviewer points out, we show that you get similar alignment with both SUs and MUs (for example, Monkey J was MUs and the other monkeys were SUs), which is consistent with the observations of Trautmann et al *Neuron* 2019. Overall, we feel that the reviewer provides a nice summary of our conclusions: similar movements are accompanied by preserved latent dynamics due to evolutionarily inherited similarities in brain architecture. Whether or not this was predictable before being empirically demonstrated may vary across individuals, but we are glad that our data seem to have convinced this reviewer about our conclusions.*

We have now stated more clearly at the beginning of the Methods that all the analyses are based on concatenated trials. We have also modified the Discussion to better elaborate the relationship between biomechanical constraints and the preservation of latent dynamics.

3. How do the observations scale with the complexity of the task? The added random-movements experiment starts to address this, but what would it look like in a cortex (or other brain region) where encoding is more sensitive, where $MU \neq SU$, where behavioral or sensory space is not 2-3 dimensional?

RNNs are not a terribly useful comparison for cortex or brains in general. Lacking this, it is difficult to know what to extrapolate from the results to other regions.

We are glad the reviewer feels that the random movement task helps to assess how the preservation of latent dynamics relates to behavioural complexity. Based on our observation that increasing the number of conditions from ~10 to 28 did not quantitatively affect the quality of the alignment, we would not expect the degree of preservation of motor cortical dynamics to be affected by the number of movements. As we discuss in the paper, we expect that there could be differences in the amount of preservation for different brain structures, e.g. in regions with potentially more sparse encoding of information or cognitive regions that are less directly connected to the ongoing production of behaviour. For example, animals may generate different dynamics while solving the same cognitive task if they are using a different strategy. Indeed, Latimer and Freedman (*Nature Comms* 2023) showed that learning experience influenced how monkeys solved the same decision-making task, leading to differences in their underlying activity. Yet, ultimately, we think any potential lack of preservation of latent dynamics will actually reflect individual differences in the “cognitive” or emotional states and/or the behavioural strategies used to solve these tasks. Implying that, if all things could be equal including these more abstract internal states and strategies, the neural population latent dynamics would be appropriately preserved.

We have now clarified this in the Discussion..

4. I now note that the primate data have appeared before in 6 publications. I know that this may be standard practice in this field, but should it be?

We have indeed used the monkey datasets across several publications, but always to answer different scientific questions on learning, the generation of behaviour or the relationship among various electrophysiological signals. We personally feel it is a good ethical practice relating to the “3Rs” principles, as it reduces the number of animals used in research. We also want to note that the majority of these datasets are publicly available so other people in the community can use them to pursue other questions.

In conclusion, the paper has been very much improved, should and could appear as it is somewhere, but whether it is sufficiently compelling to appear in Nature is unclear to me.

We are glad that the reviewer thinks that the paper has been improved since the original submission, and appreciate their thoughtful comments on our work.